# A central role of IKK2 and TPL2 in JNK activation and viral B-cell transformation

Stefanie Voigt[1], Kai R. Sterz [1], Fabian Giehler[1,2], Anne-Wiebe Mohr [1,2], Joanna B. Wilson [3], Andreas Moosmann [1,2,4] & Arnd Kieser [1,2]*

IκB kinase 2 (IKK2) is well known for its pivotal role as a mediator of the canonical NF-κB pathway, which has important functions in inflammation and immunity, but also in cancer. Here we identify a novel and critical function of IKK2 and its co-factor NEMO in the activation of oncogenic c-Jun N-terminal kinase (JNK) signaling, induced by the latent membrane protein 1 (LMP1) of Epstein-Barr virus (EBV). Independent of its kinase activity, the TGFβ-activated kinase 1 (TAK1) mediates LMP1 signaling complex formation, NEMO ubiquitination and subsequent IKK2 activation. The tumor progression locus 2 (TPL2) kinase is induced by LMP1 via IKK2 and transmits JNK activation signals downstream of IKK2. The IKK2-TPL2-JNK axis is specific for LMP1 and differs from TNFα, Interleukin−1 and CD40 signaling. This pathway mediates essential LMP1 survival signals in EBV-transformed human B cells and post-transplant lymphoma, and thus qualifies as a target for treatment of EBV-induced cancer.

[1] Helmholtz Centre Munich - German Research Centre for Environmental Health, Research Unit Gene Vectors, Marchioninistrasse 25, 81377 Munich, Germany. [2] German Centre for Infection Research (DZIF), Partner Site Munich, Munich, Germany. [3] School of Life Sciences, College of Medical, Veterinary and Life Sciences, University of Glasgow, Glasgow G12 8QQ, UK. [4] Faculty of Medicine, Ludwig-Maximilians-University Munich, Grosshadern, Marchioninistrasse 25, 81377 Munich, Germany. *email: a.kieser@helmholtz-muenchen.de

The latent membrane protein 1 (LMP1) serves as the prototype of a herpesviral oncoprotein. It is expressed by Epstein-Barr virus (EBV), a γ-herpesvirus causing lymphoproliferative and malignant diseases in humans[1,2]. LMP1 critically contributes to the pathogenesis of B-cell-derived infectious mononucleosis, Hodgkin's lymphoma and fatal posttransplant lymphoproliferative disease (PTLD), as well as playing an important role in the development of EBV-associated nasopharyngeal carcinoma[2,3]. B cell transformation by the virus depends on LMP1, which mimics stimulatory signals usually provided by the CD40 receptor[4–7]. LMP1 causes oncogenic cell transformation in rodent fibroblasts and LMP1-transgenic mice develop hyperproliferation and inflammation, or malignancies, dependent on the tissue and cell type, in which the viral oncogene is expressed[8–12].

Functioning as constitutively active receptor, LMP1 deregulates cellular pathways involved in proliferation and survival signaling including NF-κB, PI3-kinase, and the MAP kinases c-Jun N-terminal kinase (JNK), ERK and p38[1]. LMP1 consists of a short N-terminal region linked to six transmembrane helices, which mediate LMP1 activation by spontaneous autoaggregation. The C-terminal cytoplasmic domain (amino acids 187–386) harbors the C-terminal activator regions one (CTAR1, aa 194–232) and two (CTAR2, aa 351–386), which are both required for viral cell transformation[1]. Similar to TNF receptors (TNFR) or members of the Interleukin-1/Toll-like receptor (IL-1/TLR) family, LMP1 recruits TNFR-associated factors (TRAFs). TRAF1, 2, and 3 directly bind to CTAR1, the region that induces noncanonical NF-κB, PI3-kinase, and MAPK signaling[1,13]. CTAR2 recruits TRAF6 and the TNFR-associated death domain protein (TRADD) to activate canonical NF-κB and JNK as the primary readouts of this subdomain[14–19]. Both CTAR2-induced pathways are required for proliferation and survival of EBV-transformed B cells[20,21]. TRAF6 is essential for CTAR2 function[17–19]. It possesses lysine 63-linked ubiquitin ligase activity. Accordingly, K63-linked, but also linear ubiquitin chains, are important for LMP1 downstream signaling[22,23]. The Ste20 germinal center kinase TNIK, the transforming growth factor-β-activated kinase 1 (TAK1), and the TAK1 binding proteins (TAB) 1–3 are involved in JNK and canonical NF-κB signaling downstream of TRAF6[19,24–26].

The transcription factor NF-κB is a key regulator of inflammation and immunity, which is often found deregulated in lymphoid malignancies and other cancer types[27–29]. The canonical NF-κB pathway converges with the noncanonical pathway at the IκB kinase (IKK) complex, which is also the case in LMP1 signal transduction[1,30]. The IKK complex comprises two catalytic components, IKK1 (also named IKKα) and IKK2 (IKKβ). The regulatory factor NEMO (IKKγ) is essential for the canonical NF-κB pathway and acts by binding and activating IKK2 dimers[31]. IKK2 requires phosphorylation within its activation loop to be active. TAK1 can phosphorylate IKK2 at serine 177, which is a priming event for IKK2 autophosphorylation at serine 181, resulting in IKK2 activation in IL-1 signaling[32,33]. IKK2 in turn phosphorylates IκBα, whose consequent degradation leads to the liberation and nuclear translocation of canonical NF-κB dimers containing the transcription factor p65 NF-κB[28,34]. IKK2 is essential for NF-κB induction by LMP1-CTAR2 in most situations[18,26,35]. Functions of IKK2 in LMP1-mediated or cellular JNK signaling have not been reported.

The serine/threonine kinase TPL2 (tumor progression locus 2), also known as Cot or MAP3K8, plays an important role in inflammation, immunity and tumorigenesis[36,37]. TPL2 primarily activates the ERK pathway in response to various stimuli including cytokines, growth factors or stress[37–39]. Also CD40 relies on TPL2 to activate ERK[40]. In some cell types, TPL2

contributes to the optimal activation of p38 MAPK and JNK, specifically in response to genotoxic stress or inflammatory stimuli such as TNFα or IL-1β[41–44]. TPL2 crosstalks to the NF-κB pathway[37]. In its inactive state, TPL2 protein is stabilized and inhibited by complex formation with the IκB protein p105, the precursor of p50 NF-κB. Upon agonist stimulation, p105 is phosphorylated by IKK2, which causes p105 degradation and subsequent TPL2 activation and destabilization[37]. Activation-induced p105 degradation releases Rel subunits complexed with p105[30]. Further, a non-essential role of TPL2 in NF-κB signaling by LMP1 has been suggested[26,45].

Precise knowledge of the LMP1 signaling network is a prerequisite for its exploitation as a target for the treatment of EBV-associated cancer. However, the molecular mechanisms of CTAR2 signaling in particular are not fully understood. In addition to the architecture of the CTAR2 complex itself, the contributions of signaling mediators such as IKKs, TAK1 or TPL2 to CTAR2 function are insufficiently defined or controversial[19,26,45–47]. Here we investigate the functions of IKK2 in CTAR2 signaling and discover an unexpected and critical role of this IKK isoform in JNK activation by LMP1. TPL2 transmits JNK activation signals downstream of IKK2, which is a new role for this MAP3K in cellular signaling. The IKK2-TPL2-JNK pathway mediates survival and proliferation of EBV-transformed B cells and of LMP1-expressing primary tumor cells established from PTLD biopsies. Our results extend the biological functions of IKK2 and TPL2 to a critical role in JNK activation.

## Results

**Genetic analysis of IKK2 function in LMP1-CTAR2 signaling.**
To investigate the role of IKK2 in the LMP1 signaling network we transduced wildtype and $IKK2^{-/-}$ mouse embryonic fibroblasts (MEFs) with NGFR-LMP1, a fusion protein of the extracellular and transmembrane domains of p75 NGF-receptor and the signaling domain of LMP1 (Fig. 1a). This chimera can be activated by crosslinking with NGFR-directed antibodies at the cell surface, allowing the time-resolved analysis of direct LMP1 signaling events[17,48]. NGFR-LMP1(Y384G) harboring an inactivated CTAR2 was included to define the pathways that are initiated at this subdomain. Equivalent surface expression of the NGFR-LMP1 chimeras was confirmed by flow cytometry (Fig. 1b). NGFR-LMP1 crosslinking triggered the fast response of JNK and canonical NF-κB (Fig. 1c). The Y384G mutant failed to induce both pathways, demonstrating that JNK and canonical NF-κB are activated at CTAR2 (Fig. 1c). However, NGFR-LMP1(Y384G) was still able to induce the depletion of TRAF3 from NP-40-soluble cell fractions, excluding the possibility that NGFR-LMP1 (Y384G) was simply dysfunctional (Fig. 1d). TRAF3 extinction is an early step in the activation of the noncanonical NF-κB pathway, which is triggered at CTAR1[1,30]. Taken together, MEFs expressing NGFR-LMP1 constitute a suitable system to study immediate CTAR2 signaling events leading to NF-κB and JNK activation. First, we analysed the role of IKK2 in the canonical NF-κB pathway. IKK2 was essential for NGFR-LMP1 to induce IκBα degradation (Fig. 1e) and the translocation of p65 NF-κB into the nucleus (Fig. 1f). The remaining IKK1 was apparently unable to rescue the knockout of IKK2 in the canonical pathway. These results further confirmed functionality of the experimental system and demonstrated that IKK2 is in fact indispensable for canonical NF-κB activation by CTAR2 in these cells.

**IKK2 is critical for JNK activation by LMP1.** In examining the activity of the JNK pathway in $IKK2^{-/-}$ cells we made a surprising observation. JNK activation by NGFR-LMP1 was almost completely blocked by the absence of IKK2, suggesting an

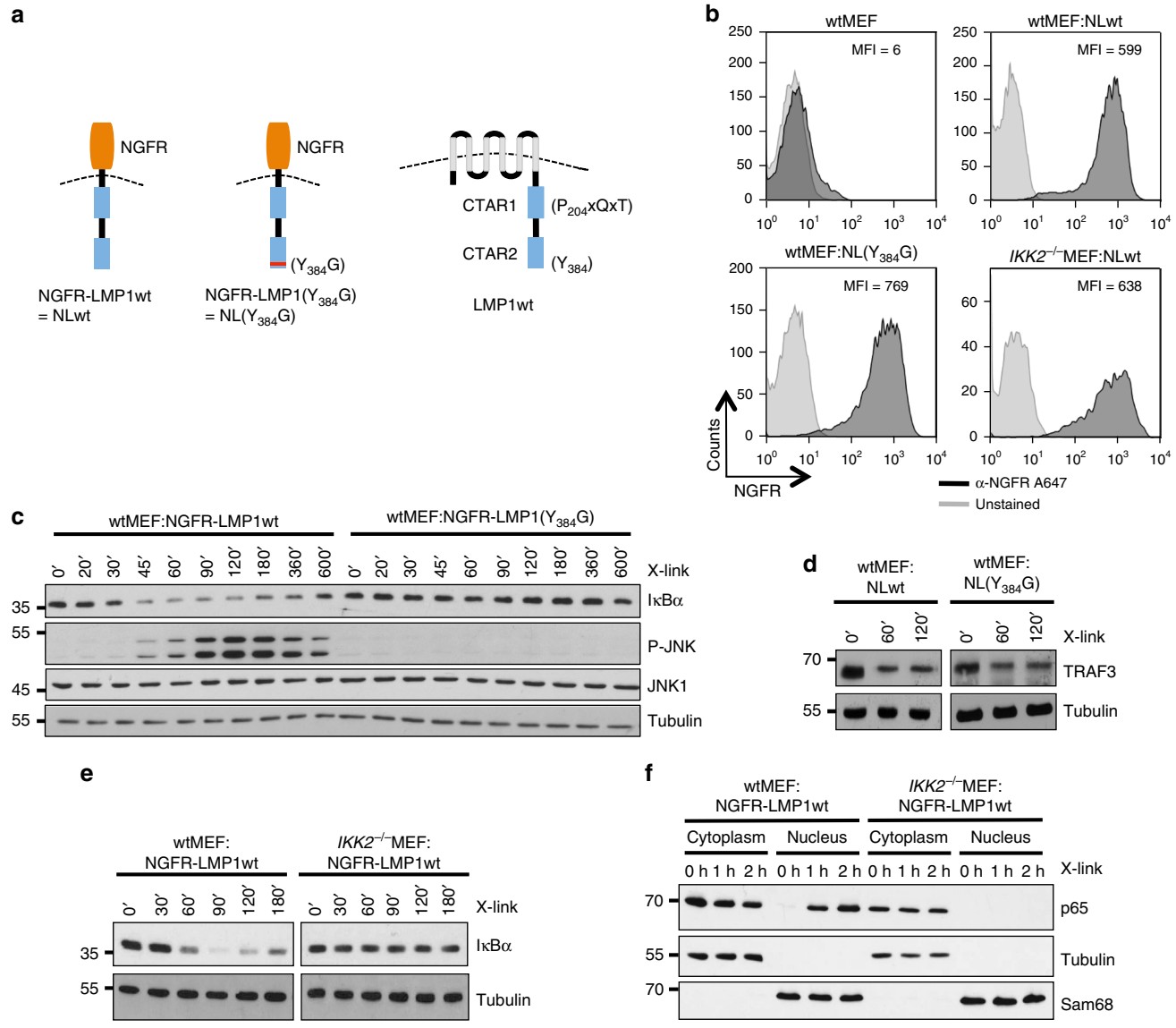

**Fig. 1 Analysis of time-resolved signal transduction events induced by the CTAR2 domain of LMP1 through IKK2. a** Schematic representation of the NGFR-LMP1 (NL) fusion constructs used in this study, compared to wildtype LMP1 (right). Y384G mutation results in a dysfunctional CTAR2 domain. Activity of NGFR-LMP1 can be triggered by NGFR-directed antibody crosslinking. **b** Wildtype and IKK2-deficient mouse embryonic fibroblasts (MEFs) were retrovirally transduced with NGFR-LMP1 or, in the case of wildtype MEFs, also with NGFR-LMP1(Y384G). Equivalent surface expression of the chimeras was confirmed by flow cytometry. MFI mean fluorescent intensity. **c** Activation of canonical NF-κB and JNK is instantly triggered at CTAR2. NGFR-LMP1 constructs were activated with NGFR antibody and a crosslinking secondary antibody for the indicated times (X-link). IκBα levels and JNK phosphorylation were detected by immunoblotting. Apparent molecular weights are given in kDa. **d** CTAR1-induced TRAF3 depletion remains unaffected by CTAR2 inactivation. Immunoblot analysis of detergent-soluble TRAF3 protein levels after crosslinking. **e** Canonical NF-κB activation by LMP1 in MEF cells depends on IKK2. **f** LMP1-induced p65 NF-κB translocation into the nucleus requires IKK2. Cytoplasmic and nuclear p65 NF-κB levels were analysed after antibody crosslinking. **b–f** The data are representative of at least two independent experiments. **c–f** For immunoblot quantification and statistics see Supplementary Table 3.

important and so far overlooked function of this IKK isoform in the LMP1 signaling pathway to JNK (Fig. 2a). Also the siRNA-mediated knockdown of IKK2 was sufficent to inhibit LMP1 signaling to JNK (Supplementary Fig. 1a). To further demonstrate that IKK activity is required for this pathway, the impact of IKK inhibitor VIII (ACHP) on JNK activation by LMP1 was tested. Indeed, ACHP reproduced the effects of IKK2 deficiency by blocking NGFR-LMP1-induced JNK1 activity (Fig. 2b). In summary, three different approaches of interference with IKK2 function verified the critical role of this IKK isoform in the JNK pathway.

LMP1 combines molecular features of TNFR and IL-1/TLR signaling, but it also uses unique mechanisms to exploit cellular signaling components, such as TRADD[1,49]. We tested, if the latter is also true for IKK2. In contrast to LMP1, TNFα was able to induce the JNK pathway in MEFs lacking IKK2, whereas both receptors relied on IKK2 to activate canonical NF-κB (Fig. 2c, compare to Fig. 1e, 2a). IL-1 activated both JNK and NF-κB in the absence of IKK2, although the response of the NF-κB pathway to IL-1 was slightly attenuated in $IKK2^{-/-}$ cells (Fig. 2d). This observation is in line with published data demonstrating that IKK1 can substitute for IKK2 in NF-κB signaling mediated by IL-1[50]. We concluded that

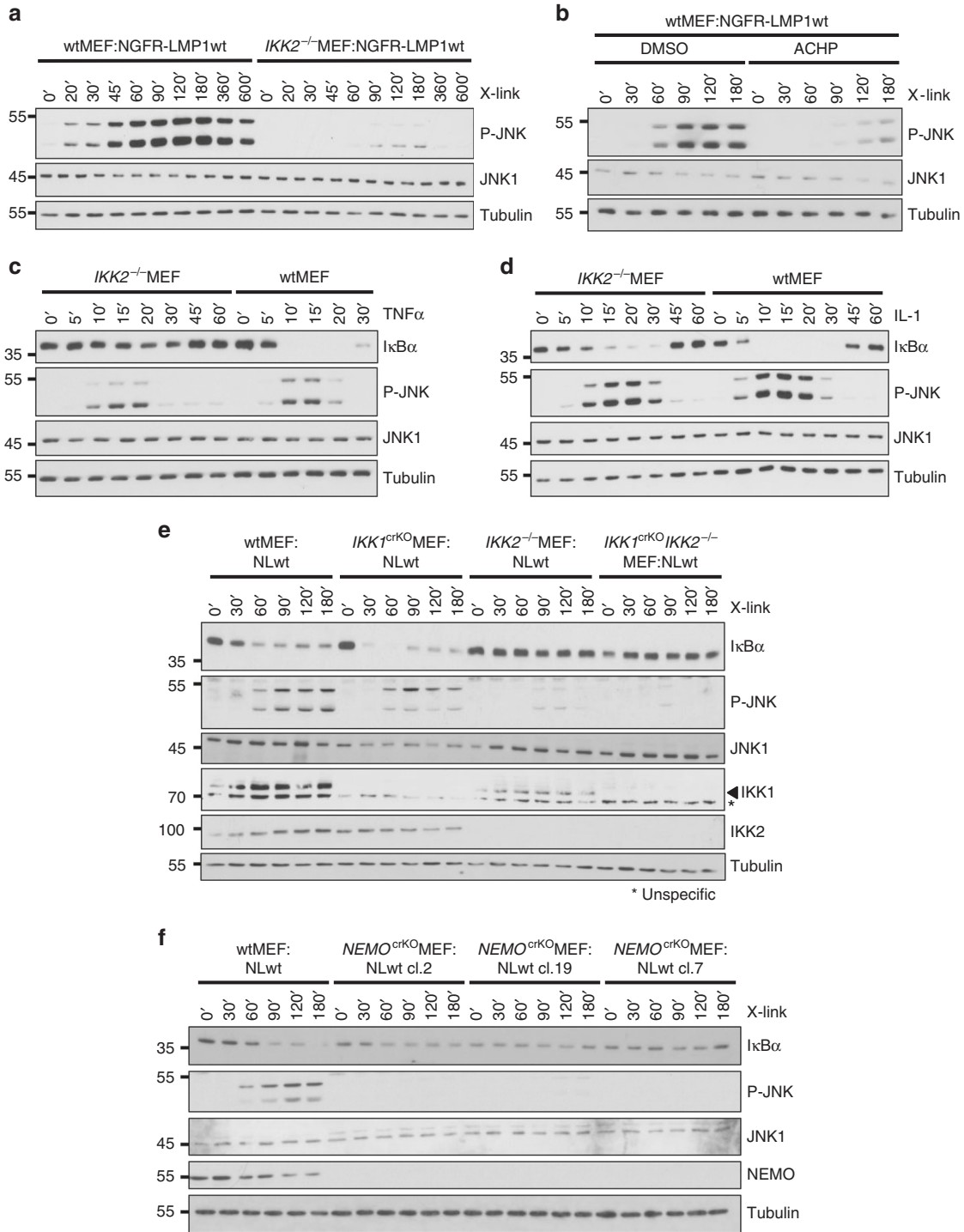

**Fig. 2 Essential functions of IKK2 and NEMO in JNK activation by LMP1. a** LMP1 fails to induce JNK after the knockout of IKK2 in MEFs. LMP1 activity was induced by antibody crosslinking and JNK activation was monitored by immunoblotting. **b** Pharmacological inhibition of IKK activity interferes with JNK activation by LMP1. NGFR-LMP1 was induced in the presence of solvent (DMSO) or 5 μM of the IKK inhibitor VIII (ACHP). **c**, **d** IKK2 has a unique role in LMP1 signaling as compared to the TNFα and IL-1 pathways. Wildtype and *IKK2⁻/⁻*MEFs were stimulated with 20 ng/ml TNFα (**c**) or 10 ng/ml IL-1 (**d**) for the indicated times. **e** IKK1 is dispensable for JNK activation by LMP1. IKK1 was targeted by CRISPR/Cas9 in wildtype MEF:NGFR-LMP1wt and *IKK2⁻/⁻*MEF: NGFR-LMP1wt cells. LMP1 activity was induced in non-targeted, *IKK1*crKOMEF:NGFR-LMP1wt (clone 69) and *IKK1*crKO*IKK2⁻/⁻*MEF:NGFR-LMP1wt (clone 8) cells. Activation of JNK and canonical NF-κB was monitored. **f** The knockout of NEMO causes a defect in JNK activation by LMP1. NEMO was inactivated in wtMEF:NGFR-LMP1wt cells by CRISPR/Cas9 technology. NGFR-LMP1 activity was induced in non-targeted cells and the *NEMO*crKO clones 2, 7, and 19. JNK and canonical NF-κB were analysed. CRISPR/Cas9 knockouts were verified by immunoblotting (this Figure) and sequencing (see Supplementary Table 2). **a–f** The data are representative of at least two independent experiments. For immunoblot quantification and statistics see Supplementary Table 3.

IKK2 has an important role in LMP1 signaling to JNK, which significantly differs from its functions in TNFα and IL-1 signaling.

**LMP1 activates JNK through NEMO, while IKK1 is dispensable**. We explored the role of the remaining IKK complex components IKK1 and NEMO in JNK activation by LMP1. The siRNA-mediated knockdown of IKK1, an essential component of the noncanonical NF-κB pathway, left JNK activation by NGFR-LMP1 unaffected, whereas the downmodulation of IKK2 strongly interfered with JNK signaling (Supplementary Fig. 1a). Next, we targeted IKK1 with the help of CRISPR/Cas9 in wildtype and *IKK2*$^{-/-}$ MEFs, both expressing NGFR-LMP1. The knockout of IKK1 had no critical effect on JNK activation (Fig. 2e and Supplementary Fig. 1b). Conversely, the knockout of NEMO blocked JNK activation induced by NGFR-LMP1 (Fig. 2f). Taken together, LMP1 uses the canonical IKK complex components NEMO and IKK2 to induce JNK.

**Differential functions of TAK1 in JNK and NF-κB signaling**. To gain deeper insight into the molecular mechanism that underlies the observed functions of IKK2 in LMP1 signaling, we investigated the role of the IKK2 kinase TAK1 in this process. First, we tested if inhibition of TAK1 kinase activity by the irreversible TAK1-specific inhibitor (5Z)−7-oxozeaenol (TAK1-IH) had an effect on JNK or canonical NF-κB activation by LMP1. Although our data have shown that IKK2 is essential for both pathways, only JNK activation was down-regulated by TAK1 inhibition (Fig. 3a). This result raised the possibility that TAK1 has no function in IKK2 activation by LMP1. To address this point, we targeted TAK1 in HEK293 cells and tested the effects of TAK1 deficiency versus inhibition of its enzymatic activity on IKK2 and JNK1 activation in parallel kinase activity assays (Fig. 3b, c, respectively). Surprisingly, the knockout of TAK1 blocked both IKK2 and JNK1 activation, whereas inhibition of TAK1 kinase activity selectively interfered with JNK1 activation. Moreover, only the complete absence of TAK1 protein was sufficient to abrogate LMP1-induced phosphorylation of IKK2 at serines 177 and 181, the critical step in IKK2 activation (Fig. 3b). Supporting IKK2 autophosphorylation rather than TAK1-mediated phosphorylation as the mechanism of IKK2 activation, the IKK kinase inhibitor ACHP blocked serine 177/181 phosphorylation, whereas the TAK1 inhibitor had no effect (Fig. 3d). These data indicated that (i) the TAK1 protein, but not its kinase activity, is essential for IKK2 activation and consequently IKK2 downstream signaling to JNK and NF-κB, (ii) TAK1 does not function as an IKK2 kinase in LMP1 signaling, and (iii) TAK1 kinase activity has a role in JNK activation parallel to IKK2. The data clearly demonstrated, however, that the TAK1-IKK2 axis is critical for JNK activation by LMP1.

Next we asked about the molecular role of the TAK1 protein in IKK2 activation by LMP1. We have shown above that NEMO is a component of the LMP1 pathway to JNK. Ubiquitination of NEMO is an important step in IKK2 activation[28]. Indeed, LMP1 induced the K63-linked ubiquitination of NEMO, which was inhibited in the absence of TAK1 (Fig. 3e). In contrast, pharmacological TAK1 kinase inhibition had little effect on NEMO ubiquitination at an inhibitor concentration that was sufficient to block JNK activation by LMP1 (Fig. 3e, compare to Fig. 3c). This result further strengthens the conclusion that TAK1 has an important function in activation of the canonical IKK complex, which is independent of TAK1 kinase activity.

LMP1 forms a signaling complex incorporating IKK2 and both proteins can be precipitated together from lymphoblastoid cells (LCLs)[49]. A possible function of TAK1 in the IKK2 pathway may

be to mediate complex formation, which results in IKK2 activation. HA-LMP1 interacted with Flag-IKK2 immunoprecipitated from transiently transfected HEK293 cells (Fig. 3f). Co-precipitation of both proteins was also possible vice versa (Supplementary Fig. 2a). Interaction of both proteins was lost in the absence of TAK1 in *TAK1*$^{crKO}$ cells (Fig. 3f and Supplementary Fig. 2a). Furthermore, NEMO has a function in signaling complex formation, as it facilitated the LMP1-induced interaction of TAK1 with IKK2 (Supplementary Fig. 2b). The data support a mechanism of IKK2 activation, in which TAK1 protein fosters signaling complex formation leading to NEMO ubiquitination and the subsequent activation of IKK2 through IKK2 autophosphorylation.

**LMP1 activates TPL2 through IKK2**. It was important to identify the mediator of JNK activation downstream of IKK2. TPL2 was a candidate, because it can be activated via IKK2 and its overexpression induces JNK, although a role of IKK2 upstream of TPL2 in JNK signaling has never been shown[37,51,52]. IKK2 directly phosphorylates TPL2 at serine 400 upon LPS stimulation[52]. Also the expression of wildtype LMP1 or NGFR-LMP1 crosslinking induced serine 400 phosphorylation of TPL2 in HEK293 cells (Fig. 4a, b, respectively). Activated TPL2 is unstable and degrades[37]. Accodingly, degradation of endogenous TPL2 started 1 h after NGFR-LMP1 crosslinking (Fig. 4c). An important step in TPL2 activation by IKK2 is the release of TPL2 from p105[37,52]. To test if LMP1 acts through a similar molecular mechanism on TPL2, wtHEK293:NGFR-LMP1wt cells were co-transfected with Myc-tagged TPL2 and Flag-p105. The expression levels of ectopic TPL2 did not alter following NGFR-LMP1 activation, however, this may be due to TPL2 overexpression. Nevertheless, the amount of TPL2 complexed with p105 decreased from 60 min after crosslinking (Fig. 4d). Altogether this shows that LMP1 induces TPL2 phosphorylation, its release from p105 and subsequent TPL2 degradation.

To confirm whether IKK2 functions upstream of TPL2 in the LMP1 pathway, we tested if disruption of IKK2 affected TPL2 activation by LMP1. The downmodulation of IKK2 expression by siRNA and its inhibition by ACHP blocked TPL2 degradation induced by NGFR-LMP1 in MEFs (Fig. 4e, f, respectively). Also the knockout of IKK2 prevented TPL2 degradation by NGFR-LMP1, supporting the function of IKK2 as mediator of TPL2 activation in LMP1 signaling (Fig. 4g).

**TPL2 is a component of the LMP1-induced JNK pathway**. TPL2 was targeted by siRNA in MEFs to assess if TPL2 is involved in JNK activation by LMP1. Indeed, the knockdown of TPL2 impaired JNK activation by NGFR-LMP1, demonstrating an important function of TPL2 in this pathway (Fig. 5a). Treatment with the ATP-competitive TPL2 inhibitor TC-S7006 reproduced the effects obtained with TPL2 siRNA (Fig. 5b). Conversely, the downmodulation of TPL2 had no effect on LMP1-induced IκBα degradation, which argues against an essential role of this kinase in canonical NF-κB activation directly downstream of IKK2 (Fig. 5a, b). CRISPR/Cas9 technology was applied to target TPL2 in MEFs expressing NGFR-LMP1. The knockout of TPL2 also caused diminished JNK activation by LMP1, whereas IκBα degradation was not significantly affected (Fig. 5c). Next, the *TPL2* gene was targeted in HEK293 cells. Three different TPL2-deficient HEK293 clones were transfected with LMP1 wildtype or the AAA/Y384G mutant together with HA-JNK1, and JNK1 activity assays were performed. JNK activation by LMP1 was substantially reduced in all three *TPL2*$^{crKO}$ clones, supporting the results previously obtained in MEFs (Supplementary Fig. 3a). Re-expression of TPL2 rescued JNK

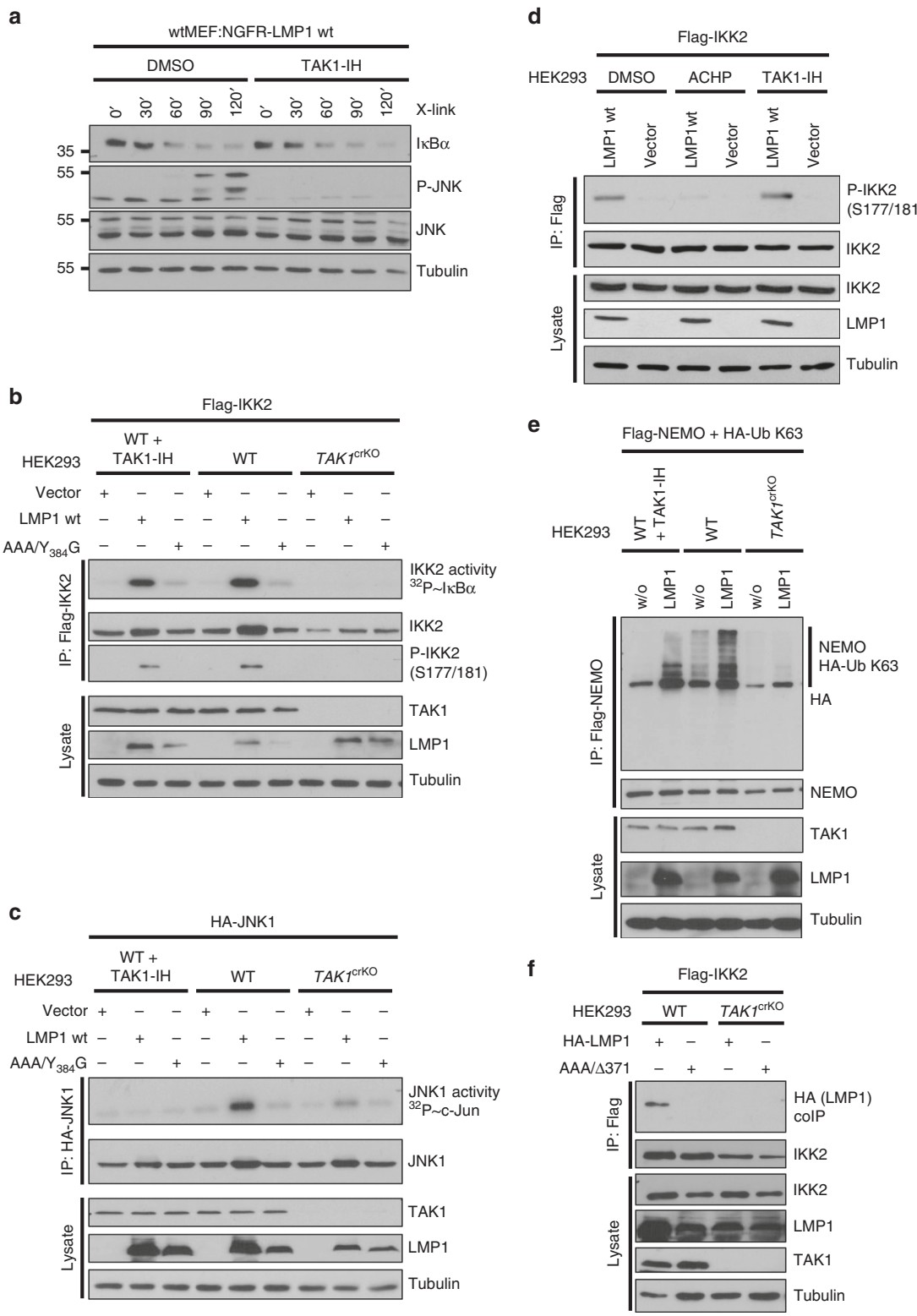

activation by LMP1 in HEK293 cells (Supplementary Fig. 3a). Taken together, LMP1 engages TPL2 downstream of IKK2 to activate the JNK pathway.

**TPL2 is dispensable for IκBα/p65 NF-κB signaling by LMP1.** Neither TPL2 deficiency nor its pharmacological inhibition were sufficient to block IκBα degradation induced by NGFR-

LMP1 (Fig. 5a–c). A role of TPL2 in NF-κB signaling by CTAR2 has previously been suggested, albeit via conflicting mechanisms[26,45]. Therefore, we analysed the effects of TPL2 deficiency on the canonical NF-κB pathway in more detail. The knockout of TPL2 did not affect NGFR-LMP1-induced translocation of p65 NF-κB into the nucleus in MEFs (Supplementary Fig. 3b). Also serine 536 phosphorylation of nuclear p65

**Fig. 3 The role of TAK1 in JNK and NF-κB signaling by LMP1. a** Pharmacological TAK1 inhibition blocks JNK, but not canonical NF-κB activation by LMP1. NGFR-LMP1 was induced by antibody crosslinking in the presence of DMSO or 500 nM of the TAK1 inhibitor (5Z)−7-oxozeaenol (TAK1-IH). **b** LMP1-induced IKK2 activity depends on the presence of TAK1 protein, but TAK1 kinase activity is dispensable. TAK1-deficient HEK293 cells (clone 3) or corresponding wildtype cells were transfected with Flag-IKK2 together with HA-LMP1, the inactive mutant HA-LMP1(AAA/Y384G), or empty vector. Where indicated, the cells were treated with 500 nM TAK1-IH for 5 h prior to cell lysis. Flag-IKK2 kinase assays were performed. IKK2 activity was measured as radioactive GST-IκBα phosphorylation (topmost panel). Phosphorylation of immunoprecipitated Flag-IKK2 at serines 177/181 was detected on immunoblots. **c** JNK activation by LMP1 requires both TAK1 protein and its kinase activity. Experiments were essentially performed as described under **c**, except that HA-JNK1 was transfected and assayed using GST-c-Jun as substrate. For kinase assay quantification and statistics see Supplementary Table 4. **d** IKK2 activation involves IKK2 autophosphorylation. HEK293 cells were transfected as indicated and treated with DMSO, IKK inhibitor ACHP or TAK1-IH. Phosphorylation of imunoprecipitated Flag-IKK2 at serines 177/181 was detected. **e** LMP1 depends on TAK1 protein, but not its kinase activity, to induce K63-ubiquitination of NEMO. HEK293 and *TAK1*crKOHEK293 cells were transfected with HA-Ubiquitin K63 and Flag-NEMO together with LMP1 wildtype or empty vector (w/o). After 24 h, cells were treated with 500 nM TAK1-IH where indicated. Flag-NEMO was immunoprecipitated (IP) using the Flag (6F7) antibody. K63-linked ubiquitination of NEMO was detected by the HA (3F10) antibody. **f** TAK1 mediates IKK2 interaction with the LMP1 complex. HEK293 wildtype and *TAK1*crKO cells were transfected with Flag-IKK2 together with HA-LMP1 wildtype or AAA/Δ371–386 lacking CTAR2. Flag-IKK2 was precipitated via the Flag (6F7) antibody and interacting LMP1 was detected by the HA (3F10) antibody. CRISPR/Cas9 knockouts were verified by immunoblotting (This Figure) and by sequencing (see Supplementary Table 2). **a–f** The data are representative of at least two independent experiments. **a–b, d–f** For immunoblot quantification and statistics see Supplementary Table 3.

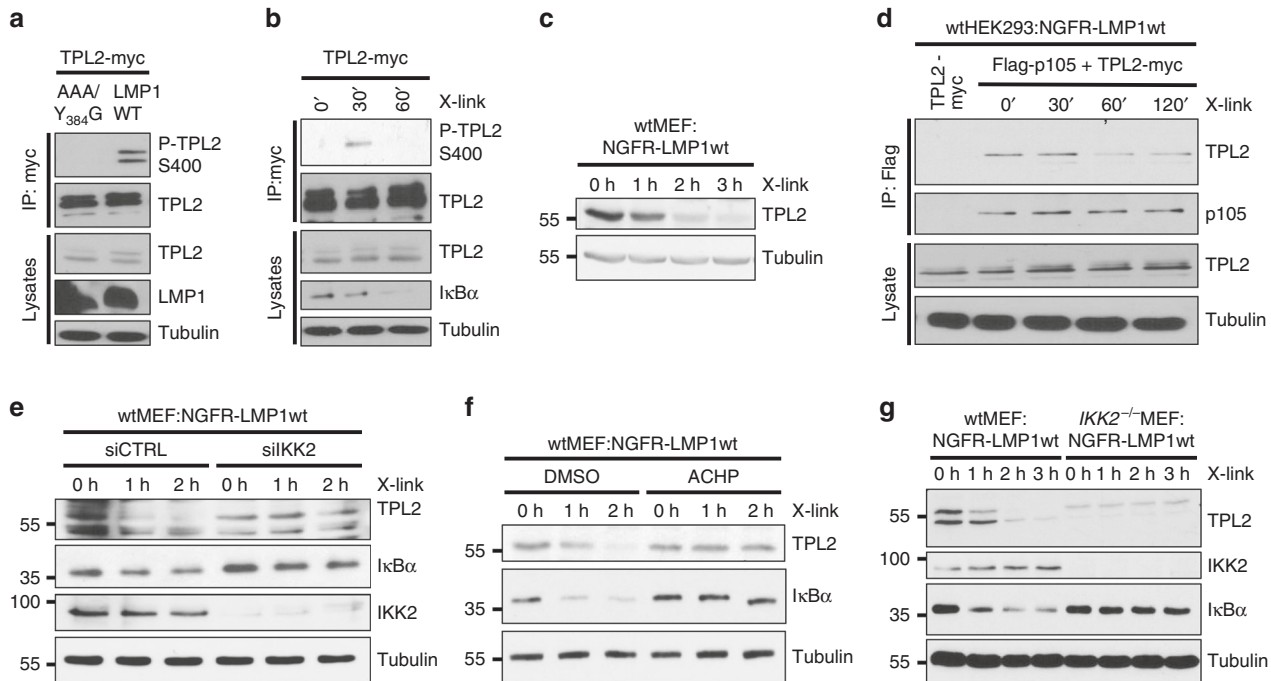

**Fig. 4 LMP1 activates TPL2 via IKK2. a**, **b** LMP1 activity induces TPL2 phosphorylation at serine 400. **a** HEK293 cells were transfected with TPL2-myc together with HA-LMP1 wildtype or inactive AAA/Y384G. Phosphorylation of immunoprecipitated TPL2-myc was detected with the P-TPL2 S400 antibody. **b** HEK293 cells, which stably express NGFR-LMP1, were transfected with Myc-tagged TPL2. NGFR-LMP1 activity was induced by antibody crosslinking 24 h after transfection. TPL2-myc was immunoprecipitated and TPL2-myc phosphorylation was detected. **c** LMP1 activity causes TPL2 degradation. Immunoblot analysis of wtMEF:NGFR-LMP1wt cells induced by antibody crosslinking. **d** TPL2 is released from its complex with p105 after LMP1 induction. HEK293:NGFR-LMP1wt cells were transfected with Flag-p105 and TPL2-myc vectors. After 24 h, NGFR-LMP1 was induced for the indicated times and Flag-p105 was precipitated via its Flag-tag. Interaction of TPL2 and p105 was analysed using the indicated antibodies. **e** The knockdown of IKK2 blocks LMP1-induced TPL2 degradation. Cells were transfected twice with IKK2 siRNA (siIKK2) or non-targeting control siRNA (siCTRL). After 24 h, NGFR-LMP1 activity was induced and TPL2, IκBα, and IKK2 levels were analysed. **f** LMP1-induced TPL2 degradation depends on IKK activity. NGFR-LMP1 was induced in MEFs by antibody crosslinking in the presence of DMSO or 5 µM ACHP. **g** LMP1 fails to induce TPL2 degradation in the absence of IKK2. NGFR-LMP1 activity was induced in wildtype and *IKK2*−/− MEFs, and cellular protein levels were analysed. **b** This experiment confirms the results of **a** and was performed once. **a**, **c–g** The data are representative of at least two independent experiments. For immunoblot quantification and statistics see Supplementary Table 3.

NF-κB remained at wildtype levels in the TPL2 knockout cells (Supplementary Fig. 3b). These results were further corroborated by the RNAi-mediated knockdown of TPL2 in MEFs, which showed no effect on p65 NF-κB translocation induced by LMP1 (Supplementary Fig. 3c). Full-length wildtype LMP1 induced the nuclear shift of p50 and p65 NF-κB in HEK293 cells, which was still the case in TPL2 knockout cells (Fig. 5d). As expected, also the CTAR1-mediated shift of noncanonical p52 and RelB remained detectable in the absence of TPL2 (Fig. 5d). Thus, TPL2 does not significantly contribute to the activation of the canonical NF-κB pathway by LMP1 at the level or upstream of NF-κB translocation.

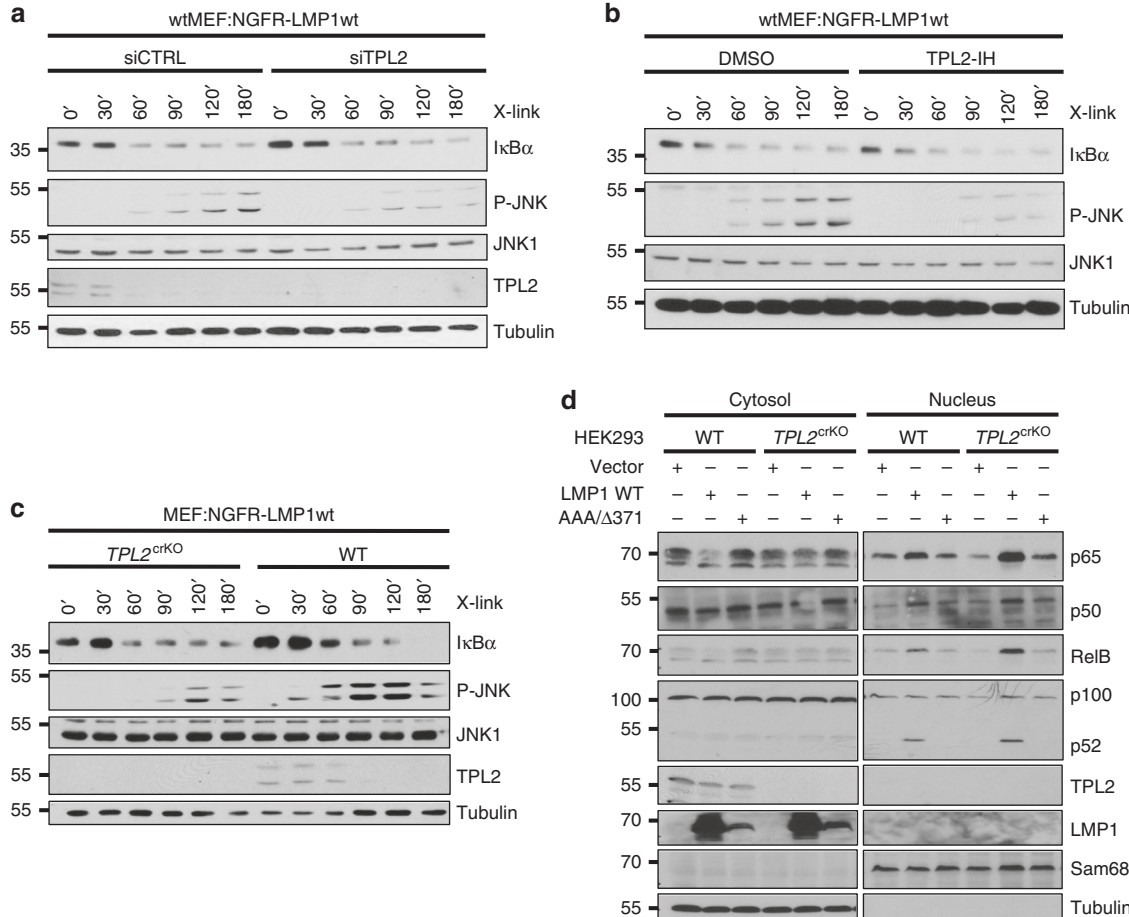

**Fig. 5 TPL2 mediates JNK activation by LMP1, but it is dispensable for NF-κB induction. a** The knockdown of TPL2 interferes with JNK activation by LMP1. Wildtype MEF:NGFR-LMP1wt cells were transfected twice with siRNA targeting TPL2 (siTPL2) or non-targeting control siRNA (siCTRL) prior to NGFR-LMP1 activation. **b** Pharmacological inhibition of TPL2 attenuates JNK activation by LMP1. Detection of JNK and NF-κB activation by NGFR-LMP1 in wtMEF:NGFR-LMP1wt cells in the presence of DMSO or 10 μM of the TPL2 inhibitor TC-S7006 (TPL2-IH). **c** The knockout of TPL2 inhibits JNK signaling by LMP1. TPL2 was targeted by CRISPR/Cas9 in wtMEF:NGFR-LMP1wt cells. NGFR-LMP1 was induced in wildtype and *TPL2*crKO (clone 8) MEFs. JNK and NF-κB activation were analysed. **d** The knockout of TPL2 in HEK293 cells does not impair canonical or non-canonical NF-κB activation by LMP1. TPL2 was inactivated in HEK293 cells by CRISPR/Cas9 technology. HEK293 wildtype and *TPL2*crKOHEK293 (clone 1) cells were transfected with HA-LMP1 wildtype, inactive AAA/Δ371–386 mutant or empty vector as indicated. Cytoplasmic and nuclear fractions were prepared and analysed by immunoblotting with the indicated antibodies. TPL2 knockouts were verified at the protein level (this Figure) and by sequencing (Supplementary Table 2). **a–d** The data are representative of at least two independent experiments. For immunoblot quantification and statistics see Supplementary Table 3.

**IKK2-TPL2 signal JNK activation by LMP1 in human B cells.**
Human B cells are the main target cells of EBV. It was therefore important to explore if this novel mechanism of JNK activation by LMP1 via the IKK2-TPL2 axis is also operative in B cells. Moreover, it was of interest to investigate if the LMP1 pathway to JNK differs from CD40 signaling. To this end, human BL41 Burkitt's lymphoma B cells expressing NGFR-LMP1 were either crosslinked with antibodies to induce LMP1 activity or stimulated with CD40 ligand in the presence or absence of inhibitors (Fig. 6a–d and Supplementary Fig. 4). LMP1 executes a similar molecular program to activate the JNK pathway in B cells as identified above in other cell types. LMP1 relied on IKK2 and TPL2 to activate JNK in BL41 cells (Fig. 6a, c, respectively). In contrast to LMP1, CD40 signaling to JNK was independent of IKK2 and TPL2 (Fig. 6b, d, respectively). The latter observation is in line with previously published data excluding a role for TPL2 in JNK signaling by CD40[40]. The same authors showed that CD40 activates the ERK pathway via TPL2 in primary murine B cells[40]. Extending this, we observed that ERK activation by both CD40 and LMP1 involves IKK2 and TPL2 in BL41 cells

(Fig. 6a–d). Finally, we addressed the role of TAK1 in CD40 and LMP1 signaling. TAK1 kinase activity was required for JNK, but not canonical NF-κB, activation by LMP1 in B cells, reflecting our previous observations in non-B cells (Supplementary Fig. 4a). In contrast, CD40 was dependent on the enzymatic activity of TAK1 to induce both NF-κB and JNK (Supplementary Fig. 4b). In summary, the mechanism of JNK activation by LMP1 is similar between B cells and non-B cells. Moreover, LMP1 and CD40 signaling differ with respect to the molecular functions of TAK1, IKK2, and TPL2.

To confirm the existence of the LMP1-induced IKK2-TPL2-JNK pathway in EBV-transformed B cells, the natural context of LMP1 function as an oncogene, we freshly infected primary human B cells with recombinant EBV, in which the LMP1 gene had been replaced by the NGFR-LMP1 chimera[53]. The resulting LCL.NGFR-LMP1.6 lymphoblastoid cells fully depend on LMP1 signals to proliferate (Fig. 6e). NGFR-LMP1 caused a robust upregulation of the JNK and canonical NF-κB pathways about 30 min after antibody crosslinking (Fig. 6f). As in non B cells and BL41 cells, LMP1 signaling to JNK was fully dependent

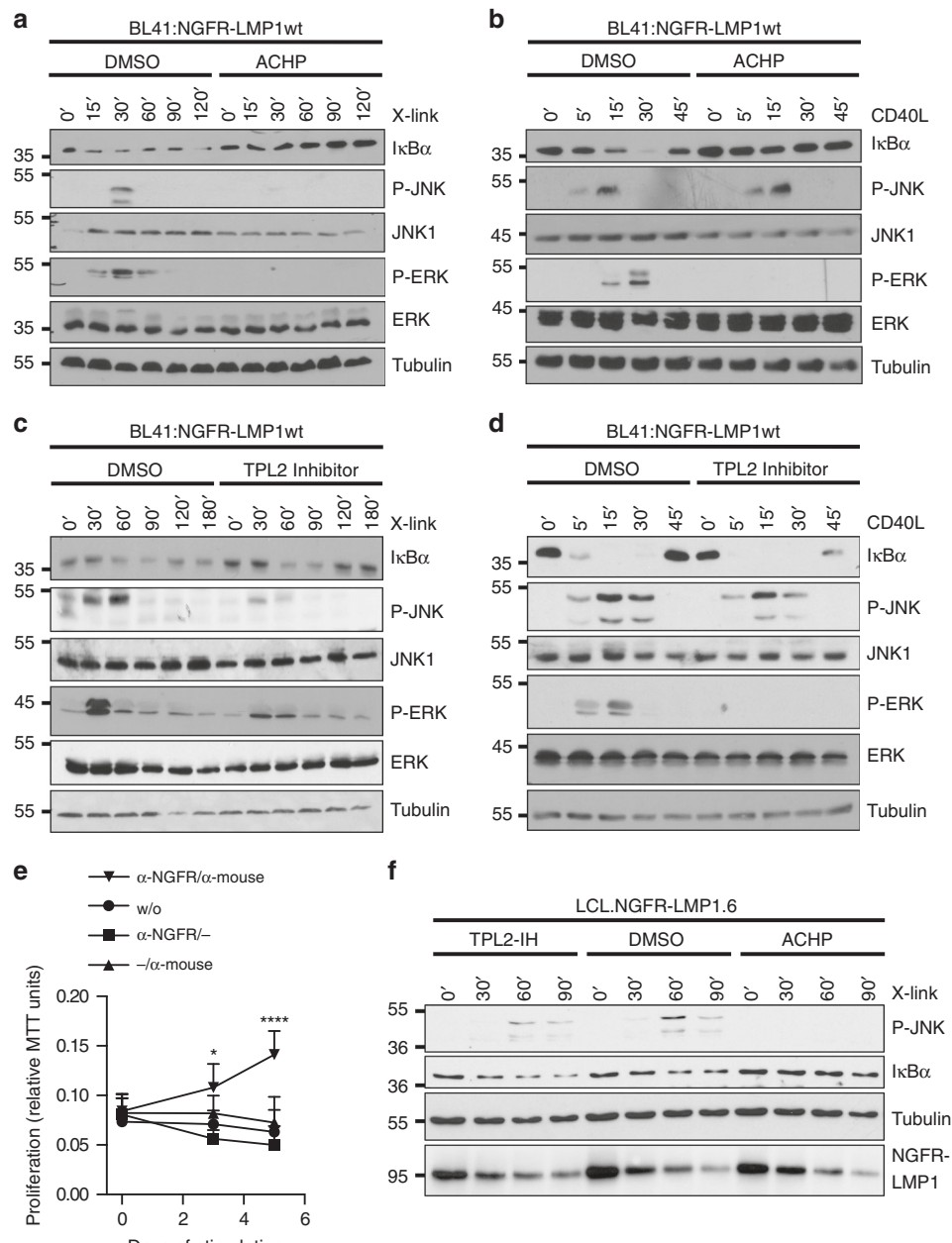

**Fig. 6 LMP1 signaling to JNK differs from CD40 and requires IKK2 and TPL2 in B cells and LCLs. a**, **b** Effects of IKK inhibition on JNK, ERK, and canonical NF-κB signaling by LMP1 (**a**) and CD40 (**b**) in human B cells. BL41:NGFR-LMP1wt cells were treated with solvent (DMSO) or 5 μM ACHP. **a** NGFR-LMP1 activity was induced by antibody crosslinking (X-link), **b** CD40 was activated with 600 ng/ml soluble CD40 ligand (CD40L) for the indicated times. JNK, ERK, and NF-κB activation was detected in total cell lysates using the indicated antibodies. **c**, **d** Effects of TPL2 inhibition. Cells were treated with DMSO or 10 μM TC-S7006 (TPL2-IH). **e** Proliferation of LCL.NGFR-LMP1.6 cells depends on LMP1 activity. LCLs were established by infection of primary human B cells with EBV expressing NGFR-LMP1 instead of wildtype LMP1. Cells were deprived of crosslinking antibodies for 7 days to fully silence the chimeric receptor. Subsequently, the cells were re-stimulated as indicated and proliferation was measured as MTT conversion. Primary antibody: α-NGFR, secondary antibody: α-mouse IgG/IgM. The data are mean values ± SD of at least four biological replicates. **f** LMP1 signaling to JNK involves IKK2 and TPL2 in EBV-transformed human B-cells. LCL.NGFR-LMP1.6 cells were activated by antibody crosslinking in the presence of DMSO, ACHP, or TPL2-IH. JNK and canonical NF-κB activity was analysed by immunoblotting. NGFR-LMP1 was detected with the LMP1 antibody 1G6-3. **e** Statistical analysis was performed with two-way ANOVA at alpha 0.05, p-values: *$p ≤ 0.05$, ****$p ≤ 0.0001$. **a–d**, **f** The data are representative of at least two independent experiments. For immunoblot quantification and statistics see Supplementary Table 3.

on IKK2 in these LCLs (Fig. 6f). Also IκB degradation induced by NGFR-LMP1 was blocked by ACHP. TPL2 inhibition caused a marked decrease in JNK activation, whereas induction of canonical NF-κB was not affected. IKK2 and TPL2 are, thus, mediators of JNK activation by LMP1 in EBV-transformed B cells, which depend on LMP1 signals.

**LCL survival is mediated by the IKK2-TPL2 pathway**. As the next step we evaluated the significance of the LMP1-induced IKK2-TPL2 pathway in cell transformation by EBV. LMP1 drives cell proliferation of LCLs and simultaneously protects cells from apoptosis by induction of NF-κB and JNK[1]. Inhibition of the canonical NF-κB pathway by expression of a dominant-negative

IκBα caused cell death in LCLs[20,54]. Consistent with these data, ACHP treatment strongly interfered with proliferation of LCLs and LCL.NGFR-LMP1.6 cells (Supplementary Fig. 5a, b, respectively).

However, as demonstrated above, IKK2 inhibition does not discriminate between NF-κB and JNK downstream signaling. To be more specific for the IKK2-TPL2 axis, we tested the effects of TPL2 inhibition on LCLs. Steady-state phosphorylation of the direct JNK target c-Jun, indicating cellular JNK activity levels, was significantly reduced already 3 h after incubation of LCL721 cells with TPL2 inhibitor (Fig. 7a). This result further confirmed that the JNK pathway is dependent on TPL2 in LCLs. Of note, JNK activity in LCLs is mediated by LMP1[5,15] (see Fig. 6f). In contrast, IκBα levels remained unaffected by TPL2 inhibition, whereas ACHP blocked canonical NF-κB and JNK in LCL721 cells (Fig. 7a). Increased apoptosis was not detected at 3–6 h after incubation with TPL2-IH, excluding cell death as an explanation for the observed effects on the JNK pathway. TPL2 inhibition efficiently retarded proliferation of the two lymphoblastoid cell lines LCL721 and LCL 1C3 as well as of LCL.NGFR-LMP1.6 cells (Fig. 7b and Supplementary Fig. 5b, respectively). TPL2-IH elicited a strong apoptotic response in LCL721 and LCL 1C3 cells after 24 and 48 h of TPL2 inhibition, demonstrating that TPL2 is required for LCL survival (Fig. 7c). This effect was specific for EBV/LMP1-transformed cells, because EBV-negative BL41 cells remained unaffected by TPL2-IH treatment (Fig. 7d). In accordance with the pro-survival function of TPL2 in lymphoblastoid cells, TPL2-IH caused an upregulation of caspase 3 activity in both LCLs, which was comparable to the effects elicited by the apoptosis inducer staurosporin (Fig. 7e).

**LMP1-positive PTLD and carcinoma depend on IKK2-TPL2-JNK.** PTLD induced by EBV constitutes a severe complication in transplant recipients. To test if the LMP1-IKK2-TPL2-JNK pathway is also effective in PTLD, primary tumor cells were established in culture from tumor biopsies of two PTLD patients, resulting in PTLD099 and PTLD880 cells. LCL877 was generated by parallel in vitro infection of primary B-cells of patient 880 with B95.8 EBV and served as LCL control for PTLD880. The EBV-encoded LMP1 genes expressed in PTLD099 and PTLD880 were sequenced. The signaling domains of the two tumor-derived LMP1 variants exhibit sequence alterations when compared to B95.8 proto-type LMP1 and when compared to each other (Supplementary Fig. 6). For instance, only LMP1 of PTLD880 carries an insertion of three 16-aa repeats. Moreover, both PTLD variants of LMP1 carry point mutations at previously described mutational hotspots (Supplementary Fig. 6 and references therein). ACHP caused a marked reduction of c-Jun phosphorylation levels in both PTLD samples as well as in the corresponding LCL877, demonstrating a downregulation of JNK activity by IKK2 inhibition in PTLD (Fig. 8a). ACHP also blocked the canonical NF-κB pathway in these tumor cells as indicated by IκBα stabilization (Fig. 8a). Notably, TPL2 inhibition interfered with JNK activity in PTLD cells, too, whereas IκBα remained unaffected (Fig. 8b). These results confirmed the relevance of the IKK2-TPL2 pathway for JNK activation even in primary PTLD tumor cells. Strikingly, this pathway is required for survival of PTLD as pharmacological IKK2 and TPL2 inhibition very efficiently blocked proliferation of PTLD cells and the corresponding LCL (Fig. 8c, d, respectively).

LMP1 is also important for nasopharyngeal carcinoma development in humans and it has oncogenic potential in rodent fibroblasts. Expressed in the epidermis of mice, LMP1 induces hyperplasia[9]. We therefore tested if TPL2 inhibition also affects the survival of LMP1-positive murine carcinoma cells, which had

been established from mice expressing an LMP1 transgene (LMP1tg) in the epidermis[9,55]. The LMP1tg-positive carcinoma line 53.234a largely depends on LMP1 for efficient proliferation and clonogenicity, whereas the LMP1tg-negative line 53.217 is independent of LMP1[55]. Indeed, TPL2 inhibition induced apoptosis in the LMP1tg-dependent 53.234a carcinoma cells, whereas 53.217 cells remained unaffected (Supplementary Fig. 7). Taken together, the IKK2-TPL2 pathway is an important mediator of JNK activation and survival of LMP1-transformed cells, observed in a variety of cell and tumor types.

**Discussion**

Here we describe an important role of IKK2 and its interactor NEMO in the activation of the JNK pathway, which is a new twist in the molecular functions and biology of the canonical IKK complex. The IKK2-dependent JNK pathway is a readout of the EBV oncoprotein LMP1 in the context of cell transformation and survival of tumor cells. LMP1 is among the most effective JNK inducers, which requires this MAPK to exert its proliferative and anti-apoptotic activity during B-cell growth transformation and tumor development by EBV[5,15,21]. Several functions in tumor-igenesis beyond NF-κB activation have been ascribed to IKK2, including the regulation of p53 stability by direct phosphorylation[56]. Our findings extend the spectrum of IKK2 to a role in oncogenic JNK signaling by engaging TPL2 as downstream mediator. TPL2 has previously been implicated in cell transformation and cancer[36]. However, a critical role of TPL2 in JNK signaling downstream of IKK2 has not been described. The IKK2-TPL2-JNK axis is thus a new signaling module regulating cell transformation.

Our results support a model of CTAR2 signaling to JNK and NF-κB, which involves IKK2 as a key player for activation of both pathways (Fig. 9). The CTAR2 signalosome includes TRAF6 as critical component proximal to the receptor[17,18]. TRADD, TAK1, TABs, TNIK, and the canonical IKK complex are further recruited[14,19,25,46,49]. We show here that TAK1 mediates IKK complex interaction with LMP1 and facilitates the ubiquitination of NEMO independent of its kinase activity. Although the exact mechanism whereby NEMO regulates IKK2 is still unknown, NEMO ubiquitination is an important step preceding IKK2 activation[28,34,56]. IKK2 requires the phosphorylation of serine residues 177 and 181 within its activation loop to be active[28,34,56]. Direct phosphorylation of IKK2 by the IKK2 kinase TAK1 is the most straightforward mechanism of IKK2 activation, as it has been demonstrated for TNFα and IL-1 signaling[32,33]. The kinase activity of TAK1, however, is not required for IKK2 induction by LMP1. Also a potential role of the alternative IKK2 kinase MEKK1 can be excluded, because LMP1 activates IKK2 in MEKK1 knockout cells[19]. How is IKK2 phosphorylation then achieved in LMP1 signaling? The requirement of TAK1 for IKK2 activation is not universal[56,57]. A discussed alternative scenario of IKK2 activation independent of an IKK2 kinase involves IKK2 binding to ubiquitinated NEMO. This interaction may then lead to conformational changes in IKK2, which facilitate the herein observed autophosphorylation of IKK2 after LMP1 activation[28,34]. Despite being dispensable for the IKK2-TPL2 axis, TAK1 kinase activity is obligatory for JNK activation by LMP1. TAK1 kinase must act towards JNK in a second pathway parallel to IKK2. Both TPL2 and TAK1 activate MAPK kinases (MKKs) that are known to regulate JNK activity[51,58]. Likely, TPL2 and TAK1 converge at the level of MKK4 and 7, which induce JNK synergistically[59]. In summary, our findings define novel functions of TAK1, NEMO, IKK2, and TPL2 in JNK signaling. They further resolve inconsistent conclusions in the literature regarding the role of TAK1 in LMP1 activation of JNK and NF-κB, which

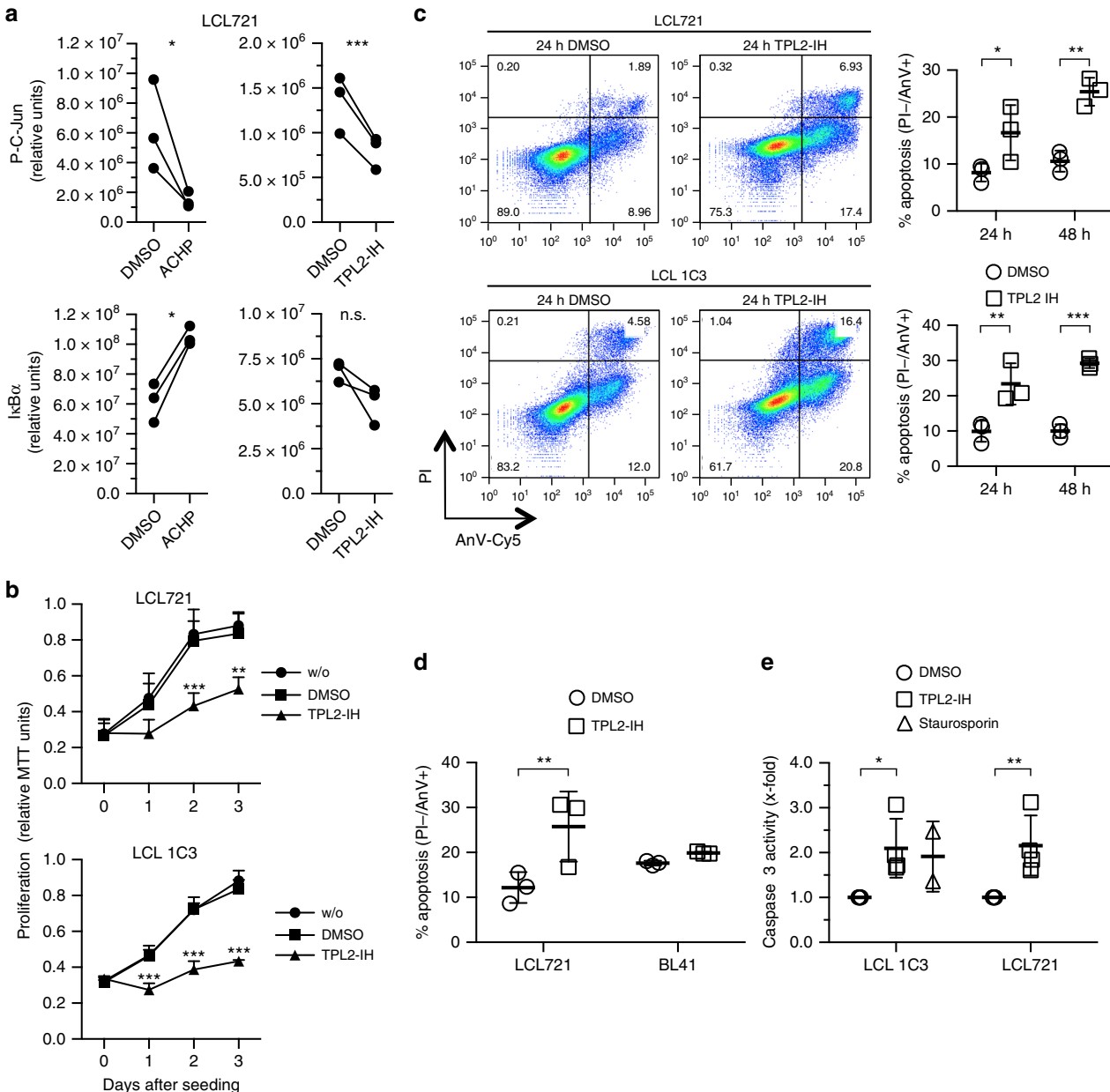

**Fig. 7 TPL2 activity is required for proliferation and survival of EBV-transformed human B cells. a** IKK2 and TPL2 inhibition causes downregulation of the JNK pathway in lymphoblastoid cell lines (LCLs). Quantitative immunoblot analysis of LCL721 cells kept in the presence of DMSO, 10 μM of ACHP (left panels) or TPL2-IH (right panels) for 3 h. The indicated antibodies were used. Paired signals standardized to total protein levels of three biological replicates are shown. Chemiluminescence values (y-axes) are relative and do not allow for direct comparison between graphs. **b** TPL2 inhibition retards proliferation of LCLs. LCL721 and LCL 1C3 cells were incubated in the absence or presence of DMSO or 10 μM TC-S7006 (TPL2-IH). Proliferation was determined as MTT conversion. Data are mean values ± SD of three biological replicates. **c** TPL2 inhibition causes apoptosis in LCLs. Cells were kept in the presence of DMSO or 10 μM TPL2-IH for 24 or 48 h, stained with propidium iodide (PI) and Annexin V-Cy5, and analysed by flow cytometry. FACS plots of one representative experiment after 24 h are shown. Percentages of AnV+/PI− cells indicating apoptotic cells are given in graphs as mean values ± SD of three biological replicates. **d** Cell death after TPL2 inhibition is specific for EBV-transformed B-cells. LCL721 and EBV-negative BL41 B-cells were grown in the presence or absence of TPL2-IH for 24 h and apoptosis rates were determined. Data are mean values ± SD of three biological replicates. **e** The activity of caspase 3 is increased in LCLs after TPL2 inhibition. Caspase 3 activity was measured after incubation of cells with DMSO or 10 μM TPL2-IH for 24 h. Staurosporin treatment for 4 h served as positive control and reference. Data represent the x-fold change ± SD of three biological replicates. **a** Statistical analysis was performed with the ratio paired *T*-test (two-tailed). **b–e** Statistical analysis was performed with two-way ANOVA at alpha 0.05. **a–e** *p*-values: *$p \leq 0.05$, **$p \leq 0.01$, ***$p \leq 0.001$, n.s. not significant.

resulted from different methods of interference with TAK1 function[19,26,46,47].

Although TPL2 is mainly involved in ERK activation by pro-inflammatory agonists, it can contribute to optimal JNK induction by TNFR1 in MEFs[41]. However, TNFR1 activates JNK independently of IKK2, whereas JNK signaling by LMP1 relies on the canonical IKK complex. How can these differences in downstream signaling between the two receptors be explained?

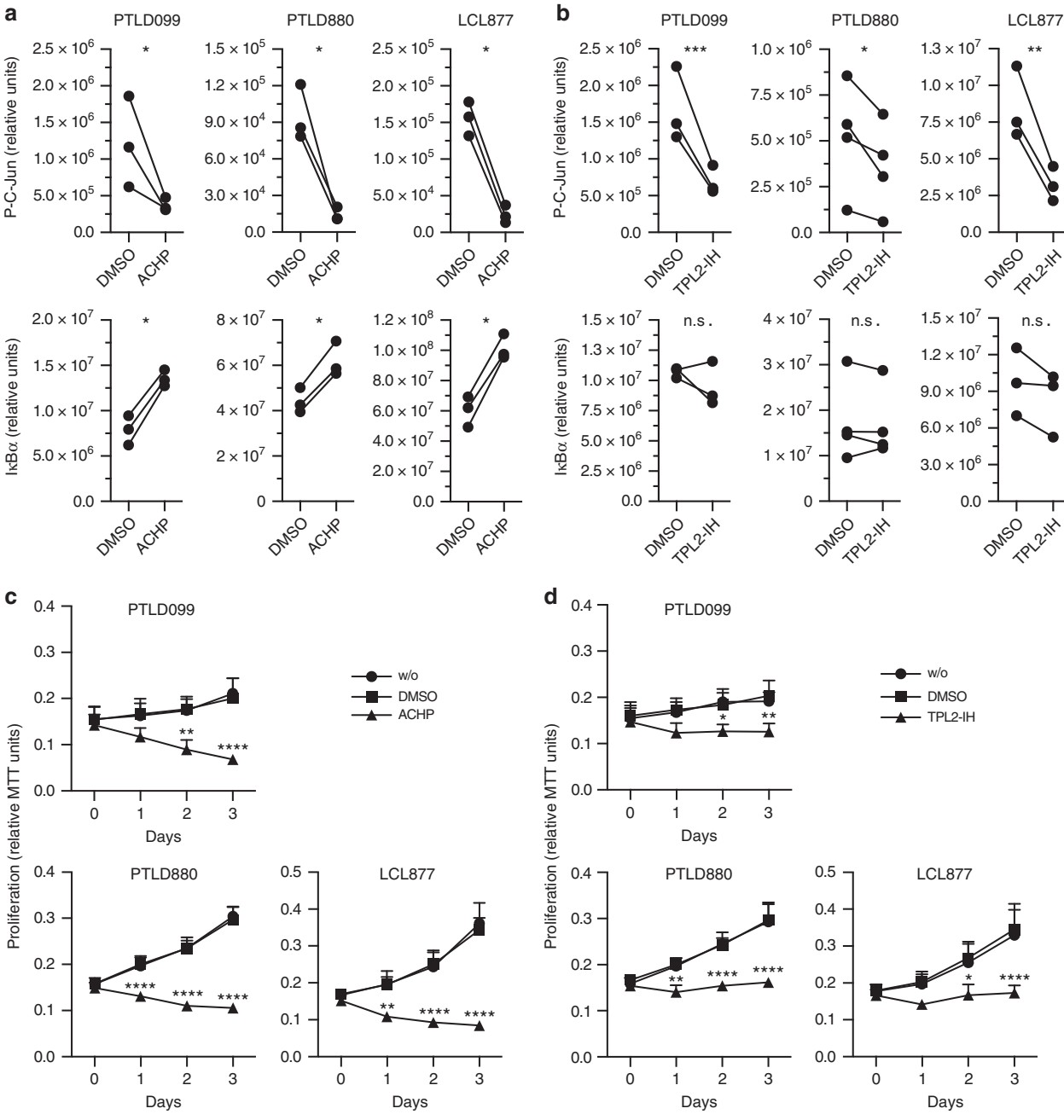

**Fig. 8 The IKK2-TPL2-JNK pathway mediates proliferation of LMP1-expressing PTLD tumor cells. a** IKK inhibition blocks the JNK and canonical NF-κB pathways in PTLD. PTLD099 and PTLD880 cells isolated from two different PTLD tumor biopsies and LCL877 corresponding to PTLD880 were incubated with DMSO or 10 μM of ACHP for 3 h. Chemiluminescence signals of immunoblots using the indicated antibodies were quantified directly. Paired signals standardized to total protein levels of three biological replicates are shown. Chemiluminescence values (*y*-axes) are relative and do not allow for direct comparison between graphs. **b** TPL2 mediates JNK, but not canonical NF-κB, activity in PTLD cells. The experiment was performed as described in **a**, except that 10 μM of TPL2 inhibitor TC-S7006 was used. Paired signals standardized to total protein levels of three (PTLD099, LCL877) or four (PTLD880) biological replicates are shown. **c** ACHP blocks proliferation of PTLD tumor cells. Cells were incubated with or without DMSO, or in the presence of 10 μM of ACHP. At the indicated times, MTT assays were performed. **d** TPL2 is required for PTLD proliferation. MTT assays in the presence of 10 μM of TPL2-IH. **c**, **d** The data are mean values ± SD of three biological replicates. **a**, **b** Statistical analysis was performed with the ratio paired *T*-test (two-tailed). **c**, **d** Statistical analysis was performed with two-way ANOVA at alpha 0.05. **a–d** *p*-values: \**p* ≤ 0.05, \*\**p* ≤ 0.01, \*\*\**p* ≤ 0.001, \*\*\*\**p* ≤ 0.0001, n.s. not significant.

TNFR1 requires TRAF2 and RIP1 to activate TPL2 and JNK[41]. In contrast, LMP1-CTAR2 signaling to JNK fully depends on TRAF6, whereas TRAF2 is largely dispensable[1]. CD40, which activates JNK independently of IKK2 in our experiments, also relies on TRAF2 for JNK signaling[60]. It is conceivable that the signaling complexes of LMP1, TNFR1 and CD40 determine

different ubiquitination processes via engagement of different TRAF ubiquitin ligases in combination with specific scaffolding proteins such as TRADD or TNIK. In line with this hypothesis, LMP1 was shown to activate IKK2 independently of the NEMO zinc finger, which binds polyubiquitin chains and is required for IKK2 activation by TNFα and CD40[35,61].

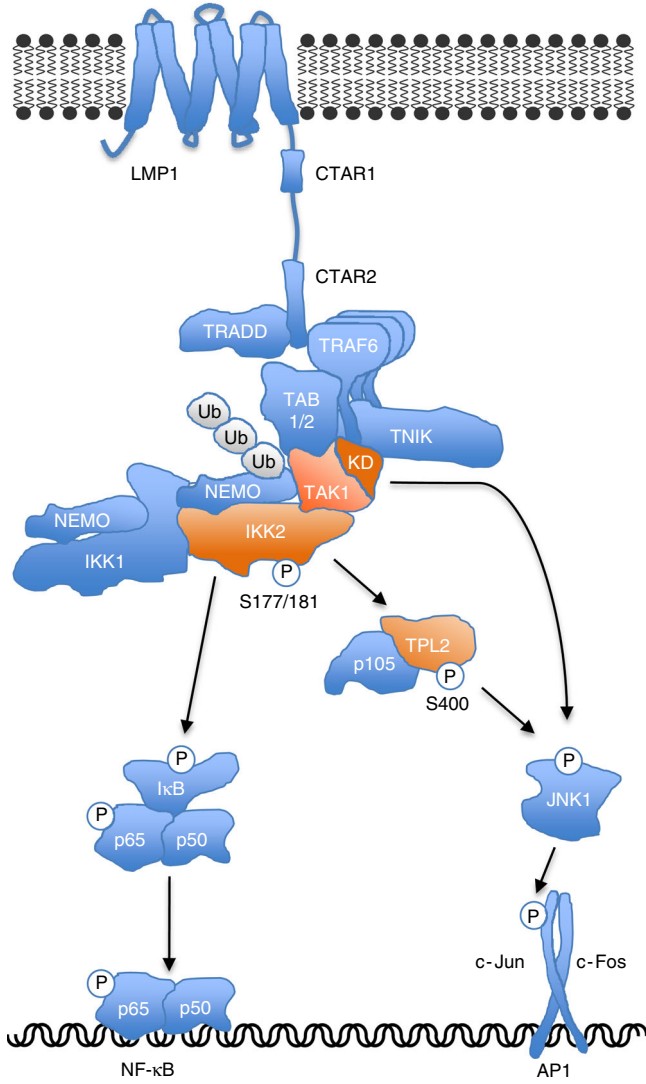

**Fig. 9 Model of IKK2, TAK1 and TPL2 functions in LMP1-CTAR2 signaling.** IKK2 acts as critical mediator of both JNK and canonical NF-κB activation by CTAR2. TPL2 transmits JNK activation signals downstream of IKK2. The TAK1 protein and NEMO are required for IKK2 recruitment to the LMP1 signalosome, thereby mediating activation of IKK2 and its downstream effectors IκBα/NF-κB and JNK. TAK1 kinase activity is dispensable for IKK2 and NF-κB activation, but has a role in JNK activation parallel to IKK2. KD, TAK1 kinase domain.

Previous reports suggested a role of TPL2 in NF-κB activation by LMP1. The overexpression of dominant-negative TPL2(K167M) showed negative effects on IκBα phosphorylation and p105 degradation induced by LMP1[45]. In another study the knockdown of TPL2 reduced transcriptional NF-κB activity, but had no effect on IκBα[26]. We showed here that LMP1-induced IκBα degradation and nuclear translocation of NF-κB remain functional even in the complete absence of TPL2. Possibly, overexpression of TPL2 (K167M) caused non-physiological effects on IκBα in the first study, for instance by the TPL2 mutant competing with endogenous signaling mediators for TRAF binding[62]. TPL2 might still be able to crosstalk to the LMP1-induced NF-κB pathway by regulating the transcriptional activity of NF-κB factors via secondary modifications or by regulating chromatin.

The LMP1-induced canonical NF-κB pathway is essential for survival of lymphoblastoid cell lines[20]. Accordingly, pharmacological inhibition of IKK activity caused cell death of LCLs. Our

results now demonstrate that IKK inhibition combines deleterious effects upon NF-κB signaling with the blockade of the JNK pathway, which is also important for LCL proliferation[21]. IKK2 could be a potential target for the treatment of EBV-induced malignancies. However, the NF-κB pathway is not only a weak point of EBV-transformed tumor cells and many lymphomas, it is also an important system in immune homeostasis[29]. Systemic IKK inhibition might therefore cause many unwanted side effects. Here we show that TPL2 inhibition interferes with survival of LCLs, PTLD tumor cells and LMP1-dependent carcinoma cells. TPL2 has previously been identified as target for anti-inflammatory drugs, because it is selectively activated by inflammatory stimuli[63]. Moreover, TPL2-deficient mice are viable, develop normally and have life spans comparable to wildtype littermates when kept under germ-free conditions[39]. Drugs targeting TPL2 should therefore have fewer systemic side effects than IKK inhibitors. On the basis of our results that TPL2 mediates oncogenic JNK signaling by LMP1 and cell survival of EBV-transformed cells, TPL2 should be considered as an attractive target for new drugs or the repurposing of existing inhibitors against EBV-induced malignancies such as PTLD in the future.

## Methods

**Cell lines and cell culture.** HEK293 cells were obtained from the German Collection of Microorganisms and Cell Cultures (DSMZ). IKK2$^{-/-}$ MEFs (provided by M. Schmidt-Supprian) were established from IKK2 knockout mice[64]. The lymphoblastoid cell line LCL 1C3 (provided by J. Mautner) was generated by infection of primary human B cells with B95.8 EBV. BL41:NGFR-LMP1wt cells (provided by J. Mautner), EBV-negative BL41 Burkitt lymphoma cells and LCL721 have been described and were taken from own laboratory stocks[65–67]. The LMP1 transgene (LMP1 tg)-positive carcinoma cell line 53.234a and corresponding LMP1-negative 53.217 carcinoma cells were established from LMP1 tg mice or LMP1-negative siblings, respectively and have been described[55]. HEK293:NGFR-LMP1 cells were generated by PolyFect (Qiagen) transfection of HEK293 cells with pSV-NGFR-LMP1 in combination with a hygromycin-resistance vector and subsequent hygromycin selection. All mouse embryonic fibroblasts, derivatives thereof, and murine carcinoma cells were cultured at 37 °C and 5% CO$_2$ in Dulbecco's modified eagle medium (DMEM) supplemented with 10% fetal calf serum (FCS) and penicillin/streptomycin. HEK293 cells, HEK293 derivatives, and B cells were kept in RPMI-1640 medium plus 10% FCS and penicillin/streptomycin. BL41:NGFR-LMP1wt cells were grown in the presence of 1 μg/ml puromycin and HEK293:NGFR-LMP1 cells in the presence of 100 μg/ml hygromycin. Cells were stimulated with the indicated concentrations of recombinant human TNFα (Applichem), murine interleukin-1 α (Tebu-Bio) or human recombinant soluble CD40 ligand (Source BioScience). The inhibitors TC-S7006 (TPL2-IH, Biotechne), (5Z)−7-oxozeaenol (TAK1-IH, Millipore), and ACHP (IKK Inhibitor VIII, Millipore) were dissolved in DMSO and used as indicated.

**Establishment of PTLD cells and LCLs.** PTLD099 and PTLD880 cells were established from anonymised tumor biopsies of two adult patients with cytopathologically confirmed EBV-positive PTLD after allogeneic hematopoietic stem cell transplantation. Tissue was disintegrated with scalpels in PBS and passed through a 100 μm Nylon cell strainer. Cells were washed and cultured in RPMI medium supplemented with 10% FCS and penicillin/streptomycin in the presence of 1 μg/ml of cyclosporine A (96-well plate, $5 \times 10^5$ cells per well in a volume of 200 μl). Half of the culture supernatant was replaced every 5–7 days by fresh medium (containing cyclosporine A for the first 4 weeks). When global cell proliferation was observed, cultures were gradually expanded until stable cell proliferation in a volume of 10 ml was achieved. At this point (day 90 for PTLD099, day 62 for PTLD880), cells were cryoconserved or further used in experiments. The LMP1 genes of PTLD099 and PTLD880 were amplified by PCR and the signaling domains were sequenced. Primer sequences are given in Supplementary Table 1. Lymphoblastoid cell line LCL877 was derived from primary cells of the same PTLD biopsy that gave rise to PTLD880, but was infected with EBV laboratory strain B95.8. Cells obtained as described above were plated in medium with cyclosporine A containing 10 μl/well of filtered (0.7 μm) supernatant from EBV-producing cell line B95.8. Cells were further cultivated and expanded as described above for PTLD cell lines.

LCL.NGFR-LMP1.6 cells were established by infection and conditional transformation of peripheral blood B cells of an adult EBV-negative donor with recombinant maxi-EBV 2264.19, carrying NGFR-LMP1 instead of wildtype LMP1[53]. Initial outgrowth of infected B cells was supported by plating PBMCs on top of an adherent layer of irradiated LL8 mouse fibroblasts expressing human CD40L[68]. At day 14, the cells were removed from the feeder layer and since then continuously cultivated in the presence of crosslinking antibodies (see NGFR-

LMP1 crosslinking) to maintain LMP1 signals and proliferation. After 8 weeks, the culture expanded to approximately $10^6$ cells and was used for experiments.

**Ethics**. We complied with all relevant ethical regulations for work with human participants. Anonymised human PTLD biopsies and blood from a healthy human donor were obtained with informed consent as approved by the Institutional Review Board (Ethics Commission of the Faculty of Medicine of the Ludwig-Maximilians-University Munich, project no. 071–06–075–06).

**Plasmids**. The plasmids pCMV-HA-LMP1 wildtype, pCMV-HA-LMP1(AAA/Δ371–386) harboring a P204xQxT to AxAxA mutation within CTAR1 and lacking the 16 C-terminal amino acids of CTAR2, pCMV-HA-LMP1(AAA/Y384G), pSV-LMP1, pSV-LMP1(Y384G), pcDNA3-Flag-IKK2, and pRK5-HA-JNK1 have been described[16,49]. The vector pSV-NGFR-LMP1 encoding a fusion protein of aa 1–279 of human low affinity p75 NGF-receptor and aa 196–386 of LMP1 has been described[17,48]. pCMV5-TPL2wt.MT (provided by C. Patriotis) and pcDNA3-Flag-p105 (provided by D. Krappmann) have been described[69,70]. The vector pEF4C-3xFlag-IKKγwt (NEMO) was a kind gift of D. Krappmann. pRK5-HA-Ubiquitin K63 (all lysines mutated to arginines except of K63) was obtained from Addgene and has been described[71].

**Retroviral transduction**. NGFR-LMP1 wildtype and NGFR-LMP1(Y384G) were subcloned from pSV-NGFR-LMP1 into the retroviral vector pSF91-IRES-GFP-WPRE (provided by C. Baum)[72]. For virus production, phoenix-gp cells were transfected with pSF91-NGFR-LMP1-IRES-GFP-WPRE, gag-pol vector and pEcoEnv expressing ecotropic Env protein as described[21]. MEFs were infected and sorted for low and comparable GFP expression levels using a MoFlo cell sorter (Beckman Coulter). NGFR-LMP1 expression at the cell surface of the resulting bulk cultures was analysed by staining with Alexa647-conjugated NGFR antibody (#557714, BD Pharmingen) and subsequent flow cytometry using a FACS Calibur flow cytometer (Becton Dickinson). Data processing was performed with FlowJo software.

**CRISPR/Cas9 gene targeting**. U6gRNA-Cas9-2A-GFP gene targeting vectors were obtained from Sigma-Aldrich and expressed Cas9, GFP and the following gRNAs: murine MM0000145296 (*IKK1*), MM0000125118 (*NEMO*), and MM0000226028 (*TPL2*), human HS0000099337 (*NEMO*), HS0000206639 (*TPL2*), and HS0000272768 (*TAK1*). Cells were co-transfected with targeting vector and a puromycin selection plasmid using PolyFect (Qiagen). Twenty-four hours after transfection, cells were seeded in different dilutions into 15 cm cell culture plates in the presence of 1.5 µg/ml puromycin. After 30 h, selection pressure was removed by washing with PBS and the cells were cultured for 2–3 weeks in standard cell culture medium. GFP-negative single clones, which had lost the targeting vector, were propagated and tested for successful gene targeting by immunoblotting and sequencing. For the latter, genomic DNA of the targeted region was amplified by PCR and sequenced (Results, see Supplementary Table 2). Primer sequences are provided in Supplementary Table 1. Gene loci targeted by CRISPR/Cas9 in cell culture were named crKO to distinguish them from classical gene knockouts by homologous recombination.

**RNA interference**. Mouse embryonic fibroblasts were transfected twice within 24 h, each time with 100 nM of ON TARGETplus SMART-pool (pool of 4 siRNAs) TPL2 siRNA L-040683-00, IKK1 siRNA L-041014-00, IKK2 siRNA L-040630-00 as indicated, or non-targeting control siRNA using the Dharmafect 1 transfection reagent (all: Dharmacon) according to the manufacturer´s protocol. Cells were stimulated and further analysed as indicated 1 day after the last transfection with siRNA.

**NGFR-LMP1 crosslinking and inhibitors**. For crosslinking experiments, $5 \times 10^5$ cells were seeded in 6 cm dishes or $1.5 \times 10^6$ cells in 10 cm dishes and starved overnight in FCS-free medium. The cells were incubated for 45 min at 37 °C with 1 µg/ml of anti-NGFR HB8737 primary antibody (ATCC). NGFR-LMP1 activity was induced by crosslinking with 10 µg/ml of anti-mouse IgG/IgM secondary antibody (#115-005-068, Dianova) for the indicated times. BL41:NGFR-LMP1 and LCL.NGFR-LMP1.6 cells were crosslinked in suspension. TPL2 and TAK1 inhibitors were added 3–5 h and the IKK inhibitor 30 min prior to antibody incubation and were present throughout crosslinking.

**Immunoblotting and quantification, immunoprecipitation**. For immunoblotting and immunoprecipitation cells were lysed on ice in NP-40 lysis buffer (50 mM HEPES pH 7.5, 150 mM NaCl, 5 mM EDTA, 0.1% NP-40) supplemented with phosphatase inhibitors (0.5 mM sodium orthovanadate, 0.5 mM NaF, 0.5 mM sodium molybdate, 0.5 mM sodium pyrophosphate, 50 mM β-glycerophosphate) and complete proteinase inhibitor (Roche). Lysates were cleared by centrifugation, separated by SDS-PAGE on 12.5% SDS-PAA gels, and blotted to nitrocellulose membrane (Bio-Rad). Immuno-blots were visualized using a secondary antibody coupled to horseradish peroxidase

and ECL reagent. For quantification of ECL signals captured on X-ray films, signals were analysed by densitometry using the ImageJ software. Results are shown in Supplementary Table 3. ECL chemiluminescence signals of Figs. 7a, 8a, b were detected directly by the Vilber Fusion FX6 Edge imager and analysed by the Vilber Bio-1D software. Signals were standardized to total protein levels. Statistical analysis was performed using the GraphPad Prism 6 software (see Statistical analysis). For co-immunoprecipitations, the cells were transfected in 15 cm dishes with the indicated plasmids using PolyFect transfection. Twenty-four hours of post transfection, cells were lysed in NP-40 lysis buffer and proteins of interest were immunoprecipitated by incubation for 3 h with the indicated antibodies covalently coupled to protein G-sepharose beads. The following primary antibodies were used: Cot (TPL2) M20 (sc-720, used at 1:500 dilution), IκBα C21 (sc-371, 1:1000), IKKα (IKK1) M280 (sc-7182, 1:500), IKKα/β (IKK1/2) H470 (sc-7607, 1:500), IKKγ (NEMO) FL419 (sc-8330, 1:500), JNK1/3 C17 (sc-474, 1:500), c-Jun H79 (sc-1694, 1:500), NF-κB p50 C19 (sc-1190, 1:500), RelB C19 (sc-226, 1:500), Sam68 C20 (sc-333, 1:1000), TAK1 M579 (sc-7162, 1:500), TRAF3 C20 (sc-949, 1:500), Tubulin B-5-1-2 (sc-23948, 1:5000) (all: Santa Cruz Biotech.), ERK1/2 p44/42 (#9102, 1:1000), P-ERK1/2 Thr202/Tyr204 (#9101, 1:1000), P-JNK Thr183/Tyr185 (#9251, 1:1000), P-JNK Thr183/Tyr185 (#4668, 1:1000), P-c-Jun Ser73 (#3270, 1:1000), P-IKKα/β (IKK1/2) Ser177/181 of IKK2 (#2697, 1:1000), NF-κB p65 (#4764, 1:1000), NF-κB p100/p52 (#4882, 1:1000), P-TPL2 Ser400 (#4491, 1:500), P-p65 Ser536 (#3033, 1:1000), (all: Cell Signaling Technology), P-JNK1 + 2 + 3 Thr183 + Thr183 + Tyr221 (ab124956, Abcam, 1:1000), NF-κB p105 (PA5-17150, 1:1000) (Thermo Scientific), HA 3F10 (11867423001, 1:1000), Flag 6F7 (SAB4200071, 1:1000), Myc 9E10 (SAB4700447, 1:1000) (all: Sigma-Aldrich), and LMP1 1G6-3[25,55,73] (provided by E. Kremmer, hybridoma supernatant used at 1:5 dilution).

**Immunocomplex kinase assays and quantification**. HEK293 cells were transfected in six-well plates with 1 µg each of the indicated constructs and 1 µg of pRK5-HA-JNK1 or 0.25 µg of pcDNA3-Flag-IKK2 using PolyFect transfection. For TPL2 rescues, 0.1 µg of TPL2-myc expression vector was co-transfected in Supplementary Fig. 3a. Twenty-four hours after transfection, cells were lysed in NP-40 lysis buffer and HA-JNK1 or Flag-IKK2 were immunoprecipitated via their epitope tags. Beads were washed twice with NP-40 lysis buffer and twice with kinase reaction buffer (20 mM Tris-HCL pH 7.4, 20 mM NaCl, 10 mM MgCl$_2$, 2 µM DTT, 2 µM ATP). In vitro kinase reactions were performed in the presence of 10 µCi γ$^{32}$P-ATP and 1 µg of recombinant GST-c-Jun or GST-IκBα, respectively, for 25 min at 27 °C[74]. Samples were subjected to SDS-PAGE and autoradiography. Radioactive signals were quantified using the Typhoon FLA-7000 phosphoimager (GE Healthcare) and normalized to the amounts of immunoprecipitated kinase. Quantitative kinase assay results are shown in Supplementary Table 4.

**Nuclear shift assay**. HEK293 cells and MEFs were treated in 10 cm dishes as indicated before cell lysis. Cells were washed in PBS at 4 °C and lysed in 100 µl of swelling buffer (10 mM HEPES pH 7.9, 10 mM KCl, 0.1 mM EDTA, 0.1 mM EGTA, 1 mM DTT, 1 mM phenylmethylsulfonyl flouride) on ice for 15 min. After adding of 0.65% NP-40, the samples were incubated for 5 min at 4 °C under gentle rocking. After centrifugation for 10 min at 15,000 × g, the supernatants representing the cytosolic fraction were collected. Pellets were washed with swelling buffer and lysed in 50 µl of nuclear extraction buffer (20 mM HEPES pH 7.9, 400 mM NaCl, 1 mM EDTA, 1 mM EGTA, 1 mM DTT, and 1 mM phenylmethylsulfonyl flouride). Insoluble debris was removed by centrifugation. Tubulin (cytoplasmic fraction) and Sam68 (nuclear fraction) served as marker proteins for purity of the preparation.

**NEMO ubiquitination**. HEK293 cells were transfected in 10 cm dishes with the indicated expression vectors, pEF4C-3xFlag-IKKγwt (NEMO) and pRK5-HA-ubiquitin K63. Cells were lysed 24 h post transfection in NP-40 lysis buffer supplemented with 10 mM N-ethylmaleimide. Flag-NEMO was immunoprecipitated via its Flag-tag and K63 linked ubiquitination was analysed by immunoblotting with HA primary antibody recognizing HA-Ub K63 modification of Flag-NEMO.

**Cell proliferation and apoptosis**. Metabolic MTT (3-(4,5-dimethylthiazol-2-yl) −2,5-diphenyltetrazolium bromide) conversion by viable cells was used to monitor cell expansion as described[5]. At day 0, 10,000 cells were seeded per well of a 96-well plate in triplicates and analysed after the indicated times. Apoptosis was assayed by staining of the cells with propidium iodide (PI) and Cy5-labeled annexin V using the Annexin V-Cy5 Apoptosis Detection Kit (K103, Biovision) as described in the manufacturer's protocol. Flow cytometric analysis of PI and annexin V-Cy5 staining was performed with the FACS CantoII flow cytometer (Becton Dickinson). FACS data were analysed with FlowJo X.0.7 software. The gating strategy for apoptosis assays is shown in Supplementary Fig. 8. All cells were included into the analysis, whereas cell debris was excluded.

**Caspase activity assay**. Caspase 3 activity was monitored using the Caspase Colorimetric Substrate Set Kit (#K132, Biovision) according to the manufacturer's protocol. Cells were treated as indicated with 10 µM TC-S7006 for 24 h before cell lysis.

Caspase 3 substrate was added to the lysates and incubated for 2 h at 37 °C. Caspase 3 activity was measured as colorimetric substrate conversion at 405 nm wavelength.

**Statistical analysis**. Statistics was performed using Graphpad Prism 6 software. $T$-test (two-tailed) was performed for unpaired samples (Supplementary Tables 3, 4). Two-way ANOVA analysis was performed for grouped samples. Ratio paired $T$-test (two-tailed) was applied for paired samples (Figs. 7a, 8a, b).

**Reporting summary**. Further information on research design is available in the Nature Research Reporting Summary linked to this article.

## Data availabilty

LMP1 sequencing data have been deposited in GenBank under the accession codes MN244676 for PTLD099 and MN244677 for PTLD880. Original uncropped scans of blots and other important original data are supplied in the Source Data file.

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

## Acknowledgements
We thank Helmut Kutz and Qinmei Yang for excellent technical assistence. We further thank Marc Schmidt-Supprian and Josef Mautner for cell lines, Christopher Baum, Ted Dawson, Daniel Krappmann and Christos Patriotis for plasmids, Elisabeth Kremmer and Regina Feederle for antibodies, and Joachim Ellwart for help with fibroblast cell sorting. This work was supported by grants Ki 825/1–3 of the German Research Foundation (DFG), e: Med 031A428G by the Federal Ministry of Education and Research, and TTU 07.802 and TTU 07.809 by the German Research Center for Infection Research (DZIF) to A.K.

## Author contributions
A.K. designed and supervised the project, conducted experiments, analysed and interpreted data and wrote the manuscript, S.V. conducted experiments, analysed and interpreted data and wrote the manuscript, K.R.S. and F.G. conducted experiments, analysed and interpreted data, A.-W.M. conducted experiments, J.B.W. contributed materials and interpreted data, A.M. contributed materials, conducted experiments, and interpreted data.

## Competing interests
The authors declare no competing interests.
