## [Peer Review File · Nature Communications]

Reviewers' comments:

Reviewer #1 (Remarks to the Author):

This manuscript describes how the EBV LMP1 protein can 'remodel' cell signalling pathways to activate JNK and NF- κ B signalling. In response to cytokine signalling, activation of the IKK complex typically requires TAK1 mediated phosphorylation of IKK β . However, although LMP1 still requires TAK1 for IKK2 activation, there is apparently no longer any requirement for TAK1 kinase activity in this LMP1 mediated process. Rather, the authors propose that the most likely mechanism is the induction of a conformational change, mediated through the association of the IKK complex with TAK1 and LMP1, that results in trans autophosphorylation by IKK2 (although proof of such a mechanism is not provided). Activated IKK2 results in activation of the TPL2 kinase (through an established mechanism) and this together with TAK1 (in a manner now requiring TAK1 kinase activity) results in activation of JNK. Active JNK in turn is required for B-cell transformation by EBV. The authors propose that their data reveals TPL2 to be a novel target for the treatment of EBV associated cancer.

The experiments in this report provide a comprehensive analysis of this pathway and are well controlled, typically taking different approaches (such as gene knockouts vs shRNA vs inhibitors) to confirm a finding. The data contained here is a mixture of new findings about the effect of LMP1 on this signalling pathway (such as the kinase activity independent role of TAK1 discussed above) together with confirmation that previously established functions (such as IKK2 mediated activation of TPL2) are operating. However, it is known that different mechanisms of activating the IKK complex exist dependent upon the nature of the cell stimulus and receptor complex being used. As such, while these findings provide valuable insights into LMP1 function, they are not in themselves particularly novel from the perspective of NF- κ B/IKK signalling.

Specific comments

(1) As the authors themselves comment, the signalling model identified here may play also operate with other viral or cellular oncogenes. Demonstrating whether this is in fact the case (or not) would strengthen the conclusions of the manuscript by placing this data more firmly in the wider context of cancer biology.

(2) The authors make use of a number of gene knockout cell lines. A good experiment, not performed, is to reexpress the gene that has been deleted to demonstrate that this recovers the phenotype seen. Moreover, re-expression of kinase dead versions of these proteins (especially TAK1) would help confirm many conclusions (such as the kinase independent role of TAK1) that currently rely on inhibitors where off target effects are always a concern. Moreover, by not re-expressing mutant proteins, the authors have missed an opportunity to better define the functional domains of proteins that lead to this novel LMP1 dependent signalling module.

(3) There is an over-reliance on MEFs and HEK 293 cells as experimental tools in this manuscript. Figure 6 does look at the BL41 Burkitt's lymphoma B cell line while Figure 7 used EBV transformed cells but the experiments performed are entirely with kinase inhibitors (and so for example the kinase independent role for TAK1 proposed is not confirmed in this more relevant setting for EBV transformation). In Figure 7, only the TPL2 inhibitor is tested (with Supp Fig 4 looking at IKK2 inhibition). An experiment that confirmed that the novel mechanistic aspects of this report were occurring in the context of an appropriate EBV transformed cell would again strongly strengthen the conclusions of this report.

Minor comments

(4) In Figure 3d, the TAK1crKO cells appear to have a reduced level of NEMO expression. Is this a general effect and is it also seen when cells are treated with the TAK1 inhibitor? Is this reduction a

cause of or consequence of the reduction in NEMO ubiquitination seen in the TAK1crKO cells?

(5) The role of p105 in TPL2 activation is entirely inferred from over-expression in HEK293 cells (Fig. 4d). What is the effect of depleting p105 in other cell systems with active LMP1?

(6) In line 330 the authors discuss previous work using BAY 11-7082, which is described as an IKK/NF- κ B inhibitor. While this compound does inhibit the pathway, more recent work has shown that it is not a direct IKK inhibitor but in fact is an inhibitor of ubiquitin E2 ligases (and therefore has very widespread effects).

(7) The authors should provide evidence that the left and right panels in Fig 7a derive from the same gel and from the same western blot exposure time.

Reviewer #2 (Remarks to the Author):

Voigt et al. describe robust, interesting findings that the LMP1 oncogene of Epstein-Barr Virus (EBV) activates JNK via a formerly unknown pathway. They provide strong data that LMP1 activates TAK1 which activates IKK2 which activates TPL2 to activate JNK. They show too that the role of TAK1 in this signaling is independent of its kinase activity. They conclude their study with data indicating that this signaling by LMP1 is needed for the survival of EBV-transformed cells. There are two modifications that would strengthen their findings.

The bulk of the data presented is images of western blots. While this data appears to be "black and white", it is not quantified so that there is little/no statistical analysis of it. The authors need to quantify their results and where appropriate provide error analysis to support their conclusions.

The authors use three cell lines to examine the proliferation and survival of LMP1-positive cells in the presence of an inhibitor of TPL2. Two of these are B-cell lines that have been in culture for years; the third is a murine cell that has been propagated in vivo and/or in vitro for many passages. The authors' suggestion that inhibiting TPL2 could be therapeutically useful for EBV-associated cancers would be substantially strengthened if they test recently isolated EBV-positive tumor cells for their survival/proliferation when LMP1 and TPL2 are each independently inhibited.

NCOMMS-18-17857, Voigt et al.
Response to reviewers

We greatly appreciated the thoughtful comments of our reviewers, which helped us to improve our manuscript. We made significant changes to the manuscript and included new data to address the major concerns of the reviewers. A special focus was set on new experiments to demonstrate the relevance of the IKK2-TPL2-JNK axis for signaling and proliferation of primary B cells freshly infected with EBV and on human PTLT tumour cells. For the latter, we isolated primary post-transplant lymphoma cells (PTLD099 and PTLT880) from tumour biopsies of two different post-transplant lymphoproliferative disease (PTLD) patients and tested their response to IKK and TPL2 inhibition. Both PTLTs were EBV-positive and expressed tumour variants of LMP1 (new suppl. Figure 6). Notably, the IKK2-TPL2-JNK pathway is in fact operative in both PTLTs and required for tumour cell proliferation (new Figure 8). We think that these new results add substantial information to the manuscript and demonstrate the relevance of our findings for human tumours.

We highlighted significant changes within the text vs. the first version of the manuscript in red color. Minor changes in wording, which do not alter the meaning of sentences, were not highlighted to maintain clarity for the reviewers. Due to the inclusion of new data the text of the manuscript had to be condensed in some parts to meet the requirements of word count limits. To meet Nature publishing requirements for the presentation of graphs with experiments of $n < 10$, bar graphs have been converted to graphs showing single data points based on the same data sets as in the first version of the manuscript.

Point-by-point response:

Reviewer #1 (Remarks to the Author):

This manuscript describes how the EBV LMP1 protein can 'remodel' cell signalling pathways to activate JNK and NF- κ B signalling. In response to cytokine signalling, activation of the IKK complex typically requires TAK1 mediated phosphorylation of IKK β . However, although LMP1 still requires TAK1 for IKK2 activation, there is apparently no longer any requirement for TAK1 kinase activity in this LMP1 mediated process. Rather, the authors propose that the most likely mechanism is the induction of a conformational change, mediated through the association of the IKK complex with TAK1 and LMP1, that results in trans autophosphorylation by IKK2 (although proof of such a mechanism is not provided).

Response: To support an activation mechanism of IKK2 that fully relies on LMP1-induced IKK2 autophosphorylation independent of IKK2 phosphorylation by TAK1, we tested the effects of IKK kinase inhibition on Ser177/181 phosphorylation of IKK2 (**new Figure 3d**, new Figure Legend and main text lines 207-210). Whereas TAK1 inhibition had no effect on IKK2 phosphorylation levels, the IKK kinase inhibitor ACPH blocked LMP1-induced IKK2 phosphorylation at Ser177/181. This result strongly argues that

IKK2 phosphorylation at these residues must be IKK2-mediated autophosphorylation and that TAK1 does not phosphorylate IKK2 in LMP1 signaling.

Activated IKK2 results in activation of the TPL2 kinase (through an established mechanism) and this together with TAK1 (in a manner now requiring TAK1 kinase activity) results in activation of JNK. Active JNK in turn is required for B-cell transformation by EBV. The authors propose that their data reveals TPL2 to be a novel target for the treatment of EBV associated cancer.

The experiments in this report provide a comprehensive analysis of this pathway and are well controlled, typically taking different approaches (such as gene knockouts vs shRNA vs inhibitors) to confirm a finding. The data contained here is a mixture of new findings about the effect of LMP1 on this signalling pathway (such as the kinase activity independent role of TAK1 discussed above) together with confirmation that previously established functions (such as IKK2 mediated activation of TPL2) are operating. However, it is known that different mechanisms of activating the IKK complex exist dependent upon the nature of the cell stimulus and receptor complex being used. As such, while these findings provide valuable insights into LMP1 function, they are not in themselves particularly novel from the perspective of NF- κ B/IKK signalling.

Response: We appreciate the acknowledgement of the novelty of our findings for LMP1. However, we would like to emphasize that the data are also novel with regard to IKK signaling and function in general. To our knowledge, it has never been shown that IKK2 has the potential to act upstream of JNK in the JNK pathway, neither in cellular nor in viral signaling, and that there is a direct pathway from IKK2 to JNK. Moreover, we provide mechanistic insight into this pathway. This is a completely new aspect of IKK2 function and biology and significantly extends the molecular spectrum of IKK2 (see Introduction third paragraph and Discussion first paragraph). Also a role of TPL2 downstream of IKK2 has never been demonstrated for the JNK pathway. It is of interest in this context that this new pathway can mediate survival and proliferative signals in tumour cells.

Specific comments

(1) As the authors themselves comment, the signalling model identified here may play also operate with other viral or cellular oncogenes. Demonstrating whether this is in fact the case (or not) would strengthen the conclusions of the manuscript by placing this data more firmly in the wider context of cancer biology.

Response: We thank this reviewer for this interesting suggestion. We payed special attention to this question and tested two other viral oncogenes, which should have the potential to induce JNK according to the literature, K15 of KSHV and LMP2A of EBV. Despite several attempts and cloning of new expression vectors, we were not able to detect any significant JNK activation by K15 in our assays. In contrast, LMP2A readily activated JNK, but this pathway was independent of IKK2, because ACHP did not block JNK activation by LMP2A, contrary to LMP1. We chose, however, not to include the LMP2A data into the manuscript because they are simply negative.

(2) The authors make use of a number of gene knockout cell lines. A good experiment, not performed, is to reexpress the gene that has been deleted to demonstrate that this recovers the phenotype seen. Moreover, re-expression of kinase dead versions of these proteins (especially TAK1) would help confirm many conclusions (such as the kinase independent role of TAK1) that currently rely on inhibitors where off target effects are always a concern. Moreover, by not re-expressing mutant proteins, the authors have missed an opportunity to better define the functional domains of proteins that lead to this novel LMP1 dependent signalling module.

Response: We agree that unspecific off target effects are a concern when using inhibitors. This is why we confirmed our key observations with other methods of interference with gene/protein function whenever possible and in different cell types (siRNA, gene knockout). However, in our experiments exploring the relevance of TAK1 kinase activity for IKK activation, the TAK1 inhibitor has **no** effect at all on I κ B and IKK2 activation (Figures 3a, 3b, 3d), which excludes the problem of unspecific side effects and a subsequent misinterpretation of inhibitor data for this novel mechanism. Therefore, we chose to rather rescue TPL2 deficiency, where effects of the inhibitor are evident and important for the conclusions of our research. Re-expression of TPL2 was sufficient to enable LMP1 to activate JNK to full levels in all three TPL2-crKO lines (**new suppl. Figure 3a** and main text lines 276-277). Although the amounts of transfected TPL2 expression vector were titrated to very low levels, the exogenously expressed TPL2 was able to induce JNK to medium levels by itself. However and notably, exogenous TPL2 rescued TPL2 deficiency in TPL2-crKO cells, because LMP1 induced JNK to levels of WT cells after TPL2 re-expression (new suppl. Figure 3a). We also tested a kinase-negative mutant of TPL2 in these experiments, which did not rescue JNK signaling, as expected, but significantly reduced JNK activity base levels due to yet unclear reasons, possibly by acting as a dominant-negative allele or by altering stoichiometry of signaling complexes. Therefore, we included the rescue data for TPL2 wildtype only.

(3) There is an over-reliance on MEFs and HEK 293 cells as experimental tools in this manuscript. Figure 6 does look at the BL41 Burkitt's lymphoma B cell line while Figure 7 used EBV transformed cells but the experiments performed are entirely with kinase inhibitors (and so for example the kinase independent role for TAK1 proposed is not confirmed in this more relevant setting for EBV transformation). In Figure 7, only the TPL2 inhibitor is tested (with Supp Fig 4 looking at IKK2 inhibition). An experiment that confirmed that the novel mechanistic aspects of this report were occurring in the context of an appropriate EBV transformed cell would again strongly strengthen the conclusions of this report.

Response: We put much effort into new experiments in response to this important concern by this reviewer, especially because also reviewer #2 raised a similar point (see last paragraph of comments reviewer #2). Please refer to our response to reviewer #2 for a more detailed description of our new experiments in appropriate EBV transformed cells and PTLT tumour cells (**new Figures 6e-f, new Figure 8, new suppl. Figure 5b**).

The reason why we used MEFs and HEK293 cells for most of the mechanistic and gene targeting studies were threefold: (i) Both cell types are readily transfectable and accessible for manipulation with siRNA, retroviruses and CRISPR/Cas9 to knockdown or

knockout genes involved in LMP1, NF- κ B and JNK signaling without killing the cells. LCLs rely on IKK2 and TPL2 (NF- κ B and JNK activity) for cell survival. They would not survive KD or KO of genes involved in NF- κ B and JNK signaling. Sufficient cells for mechanistic resp. biochemical studies would not be available. Therefore, we performed most mechanistic studies in MEFs (where also mouse gene knockout lines are available) and HEK293 first, to confirm key results in B cells, LCLs and PTLN cells later on with inhibitors. Inhibitors allow for the ad hoc inhibition of the kinases to study their functions in signaling and open a time window in which the kinases are already inactive while the cells are still viable (3-6 h of inhibition). (ii) LMP1 is known to have biological functions with regard to cell transformation in mice, and HEK293 is the preferred and well-accepted model cell system to study LMP1 signaling in the literature. (iii) MEFs and HEK293 allow biochemical experiments under gene knockdown or knockout conditions in the NF- κ B and JNK pathways.

Minor comments

(4) In Figure 3d, the TAK1crKO cells appear to have a reduced level of NEMO expression. Is this a general effect and is it also seen when cells are treated with the TAK1 inhibitor? Is this reduction a cause of or consequence of the reduction in NEMO ubiquitination seen in the TAK1crKO cells?

Response: The slightly reduced levels of transfected NEMO in the TAK1-crKO cells of experiment 3d (now Figure 3e) rather reflect marginal variations in transfection efficiencies between the cell lines in this specific experiment and were not consistent throughout all experiments. To demonstrate that the knockout of TAK1 does not reduce NEMO expression levels, we analysed endogenous NEMO expression in TAK1-crKO versus HEK293 wildtype cells. As shown in the Figure below, endogenous NEMO levels were not reduced in TAK1-crKO cells.

Immunoblot analysis of endogenous NEMO expression in HEK293 cells with CRISPR/Cas9-mediated knockout of TAK1 (TAK1-crKO).

(5) The role of p105 in TPL2 activation is entirely inferred from over-expression in HEK293 cells (Fig. 4d). What is the effect of depleting p105 in other cell systems with active LMP1?

Response: TPL2 forms a stoichiometric association with p105 and is quantitatively associated with p105 in unstimulated cells¹⁻³. Vice versa, only a small fraction of p105 is associated with TPL2⁴. Therefore, it is difficult to detect this interaction with endogenous proteins and the use of overexpression to study this complex is standard in the field². We were unable to detect p105-TPL2 interaction with endogenous proteins in the cell systems used in our study.

It is well established in the literature that p105 stabilizes TPL2 and thereby facilitates its signaling capacities^{2, 5}. The depletion of p105 (NF- κ B1) per se leads to de-stabilization

of TPL2, TPL2 depletion and defective TPL2 signaling^{3,5}. Moreover, the knockout of p105 has massive effects on canonical NF- κ B signaling, because it is the precursor of p50 NF- κ B⁶. It is thus to be expected that p105 depletion will cause instant TPL2 degradation as well as defects in canonical NF- κ B signaling. Both will be deleterious to LMP1-dependent cells. To our opinion depletion of p105 will thus not be very informative regarding the activation mechanism of TPL2 by LMP1/IKK2, because it will lead to TPL2 degradation and pleiotropic effects in the cell, which are not only related to TPL2. TPL2 release from p105 and its subsequent degradation is established in the literature as indication of TPL2 activation and we show here that this is also the case for LMP1 (Figure 4d). Moreover, we provide additional and supporting evidence that LMP1 in fact activates TPL2 (Figures 4a, b, LMP1-induced Ser 400 phosphorylation of TPL2; Figure 4c, LMP1-induced TPL2 degradation).

(6) In line 330 the authors discuss previous work using BAY 11-7082, which is described as an IKK/NF- κ B inhibitor. While this compound does inhibit the pathway, more recent work has shown that it is not a direct IKK inhibitor but in fact is an inhibitor of ubiquitin E2 ligases (and therefore has very widespread effects).

Response: We thank reviewer #1 for this hint. We deleted BAY 11-7082 from the sentence, lines 334-335 of revised manuscript, but kept the key message that inhibition of the canonical NF- κ B pathway causes cell death of LCLs, because this has also been shown with dominant-negative I κ B (see references 20, 54).

(7) The authors should provide evidence that the left and right panels in Fig 7a derive from the same gel and from the same western blot exposure time.

Response: Figure 7a has been replaced by new quantitative data in LCL721 cells in the revised manuscript (see also response to reviewer #2). However, to demonstrate that the left and right panels of former Figure 7a were generated from the same blots we show here the original scans of the P-c-Jun and I κ B immunoblots as representatives. The 24 h time point had been excluded because the cells already underwent apoptosis at this time point (see Figure 7c) and signaling data were considered not accurate any more after 24 h of TPL2-IH incubation.

original scan of P-c-Jun blot

original scan of I κ B blot

Reviewer #2 (Remarks to the Author):

Voigt et al. describe robust, interesting findings that the LMP1 oncogene of Epstein-Barr Virus (EBV) activates JNK via a formerly unknown pathway. They provide strong data that LMP1 activates TAK1 which activates IKK2 which activates TPL2 to activate JNK. They show too that the role of TAK1 in this signaling is independent of its kinase activity. They conclude their study with data indicating that this signaling by LMP1 is needed for the survival of EBV-transformed cells. There are two modifications that would strengthen their findings.

The bulk of the data presented is images of western blots. While this data appears to be "black and white", it is not quantified so that there is little/no statistical analysis of it. The authors need to quantify their results and where appropriate provide error analysis to support their conclusions.

Response: As this reviewer already stated, most of the immunoblot data in non-B-cells is black-and-white, especially when comparing wildtype with knockout cells. Very often, many lanes in the knockouts of key figures are simply blank (because of massive effects) and cannot be quantified properly (e.g. Figures 1c, 1f, Figure 2, Figure 3a, Figures 4a, 4b, Figure 6a, 6b, 6f, former Figure 7a). Moreover, (retrospective) densitometric quantification of immunoblots (films) is not considered as an accurate method of linear quantification any more.

However, we acknowledge the concern of this reviewer that immunoblot data should be quantified and statistically analysed where appropriate and important for the conclusions. We considered immunoblot data in LCLs and PTLT cells in the presence of IKK and TPL2 inhibition as suitable and for quantification. After submission of the first version of our manuscript, our lab has gained access to a Fusion FX6 Edge imager (Vilber), which allows the direct and linear detection, analysis and quantification of chemiluminescence (ECL) signals of immunoblots. We therefore decided to perform new experiments in LCL721 and the novel PTLTs PTLT099 and PTLT880 as well as the freshly infected LCL887 (see also response to the next point of reviewer #2) with ACHP and TPL2-IH to generate a quantitative picture with statistical analysis of the effects of IKK and TPL2 inhibition on I κ B and P-c-Jun (direct JNK target indicating JNK activity) in EBV-transformed B cells and PTLT tumours. These new data are now presented in the **new Figures 7a, 8a and 8b**. Former Figure 7a has been replaced by the new quantitative data in LCL721. Altogether the data demonstrate that the IKK2-TPL2-JNK pathway is also operative in LCLs and PTLT.

The authors use three cell lines to examine the proliferation and survival of LMP1-positive cells in the presence of an inhibitor of TPL2. Two of these are B-cell lines that have been in culture for years; the third is a murine cell that has been propagated in vivo and/or in vitro for many passages. The authors' suggestion that inhibiting TPL2 could be therapeutically useful for EBV-associated cancers would be substantially strengthened if they test recently isolated EBV-positive tumor cells for their survival/proliferation when LMP1 and TPL2 are each independently inhibited.

Response: We are grateful for this comment, which has also been raised by reviewer #1

in a similar way. Therefore, we concentrated our efforts on performing a set of new experiments to address this concern in detail.

We generated new LCLs by freshly infecting primary human B cells with recombinant EBV expressing NGFR-LMP1 instead of LMP1 wildtype (LCL.NGFR-LMP1.6 cells). These cells were used in experiments directly after cell expansion post infection. In these cells, LMP1 activity can be switched off (inhibited) by withdrawal of crosslinking antibodies. LCL.NGFR-LMP1.6 cells are clearly dependent on antibody crosslink resp. LMP1 signals for their proliferation/survival (**new Figure 6e**). Notably, IKK and TPL2 inhibition blocks proliferation in these recently established LCLs (**new suppl. Figure 5b**). Moreover, IKK and TPL2 inhibition interfere with the NGFR-LMP1-induced JNK pathway in LCL.NGFR-LMP1.6 cells, validating the IKK2-TPL2-JNK pathway also in these LCLs, which are clearly LMP1-dependent (**new Figure 6f**, lines 318 - 329).

To demonstrate the relevance of the novel IKK2-TPL2-JNK axis for signaling and proliferation of EBV-positive tumour cells, we isolated primary post-transplant lymphoma cells (PTLD009 and PTLD880) from tumour biopsies of two different post-transplant lymphoproliferative disease (PTLD) patients and tested their response to IKK and TPL2 inhibition. Both PTLDS are EBV-positive and express tumour variants of LMP1 (**new suppl. Figure 6**). Notably, the IKK2-TPL2-JNK pathway is in fact operative in both PTLDS (**new Figures 8a and 8b**, see also above, first point of reviewer #2) and required for tumour cell proliferation (**new Figures 8c and 8d**, lines 359 - 379). In addition, an LCL was established by infection of primary B cells of patient PTLD880 with B95.8 EBV. The resulting LCL887 corresponds to PTLD880 and shows a similar dependence on the IKK-TPL2-JNK pathway as the other LCLs and PTLD cells (**new Figures 8a - 8e**).

Altogether, these new data demonstrate that the LMP1-IKK-TPL2-JNK axis is operative in EBV-transformed human B cells and EBV-positive human tumour cells and is essential for their proliferation/survival.

References

1. Belich, M. P., Salmeron, A., Johnston, L. H. & Ley, S. C. TPL-2 kinase regulates the proteolysis of the NF-kappaB-inhibitory protein NF-kappaB1 p105. *Nature* **397**, 363-368 (1999).
2. Beinke, S. *et al.* NF-kappaB1 p105 negatively regulates TPL-2 MEK kinase activity. *Mol. Cell. Biol.* **23**, 4739-4752 (2003).
3. Beinke, S. & Ley, S. C. Functions of NF-kappaB1 and NF-kappaB2 in immune cell biology. *Biochem. J.* **382**, 393-409 (2004).
4. Lang, V. *et al.* ABIN-2 forms a ternary complex with TPL-2 and NF-kappa B1 p105 and is essential for TPL-2 protein stability. *Mol. Cell. Biol.* **24**, 5235-5248 (2004).

5. Yang, H. T. *et al.* NF-kappaB1 inhibits TLR-induced IFN-beta production in macrophages through TPL-2-dependent ERK activation. *J. Immunol.* **186**, 1989-1996 (2011).
6. Sha, W. C., Liou, H. C., Tuomanen, E. I. & Baltimore, D. Targeted disruption of the p50 subunit of NF-kappa B leads to multifocal defects in immune responses. *Cell* **80**, 321-330 (1995).

Reviewers' comments:

Reviewer #1 (Remarks to the Author):

I am happy with the revised manuscript and the authors response to my original review. I have no further concerns.

Reviewer #2 (Remarks to the Author):

The authors have addressed most of the concerns in their revised manuscript but failed to address their lack of validation of their first six figures with statistical support. They argue that "(retrospective) densitometric quantification of immunoblots (films) is not considered an accurate method of linear quantification any more." They do not have to use any method they deem inaccurate; what they cannot do is to ignore the problem. The data they present in the first six figures needs to be validated with biological replicates and statistical analysis of the replicates. Only if they find that the differences they now claim are, in fact, supported by the statistical analysis, can they continue to claim them. The authors, perhaps, are also disingenuous in their response to the reviews in that they do present measurements from their blots in one case, Figure 3C (without detailing how they obtained these numbers nor providing a statistical analysis of them).

NCOMMS-18-17857A, Voigt et al.

Response to reviewers

We appreciate the reviewers' comments to the first revision of our manuscript and are particularly happy that we were able to address most of their concerns, except the lack of statistical analysis of the immunoblot data of Figures 1 to 6 (reviewer #2). To meet this concern of reviewer #2 we have now quantified all immunoblot data including biological replicates of Figures 1 to 6 and provide the results including their statistical analysis in the new supplementary Table 2. Kinase assay data were also statistically analysed and are presented in the new supplementary Table 3. Changes within the text versus the previous version (NCOMMS-18-17857A) have been highlighted in red color.

Point-by-point response:

Reviewer #1 (Remarks to the Author):

I am happy with the revised manuscript and the authors response to my original review. I have no further concerns.

Response: We greatly appreciate this comment.

Reviewer #2 (Remarks to the Author):

The authors have addressed most of the concerns in their revised manuscript but failed to address their lack of validation of their first six figures with statistical support. They argue that "(retrospective) densitometric quantification of immunoblots (films) is not considered an accurate method of linear quantification any more." They do not have to use any method they deem inaccurate; what they cannot do is to ignore the problem. The data they present in the first six figures needs to be validated with biological replicates and statistical analysis of the replicates. Only if they find that the differences they now claim are, in fact, supported by the statistical analysis, can they continue to claim them.

Response: We acknowledge the concern of reviewer #2 that it is necessary to provide quantitative data and statistical support also for the immunoblots of Figure 1 to 6 and paid considerable attention to this point. In addition to the imager-based quantification of the immunoblots of Figures 7 and 8, which we added to the previous (1st) revision of our manuscript, we have now also quantified the immunoblots of Figures 1 to 6 including all biological replicates and performed statistical analysis of them. The results of this analysis are compiled in the new supplementary Table 2. Despite our initial concern that densitometric quantification of blots (films) might be suboptimal due to a lower dynamic range, we have now quantified all existing films by densitometry using the ImageJ software. Comparing imager and densitometric data for some new experiments unrelated to this manuscript, we realised that the results obtained with these two quantification techniques are very well comparable. Therefore, we have revised our opinion regarding quantification of immunoblot photography. Quantification of immunoblots is described in "Immunoblotting and quantification, immunoprecipitation, antibodies" in the Methods section. To address the situation that

on many blots/films a lot of samples lack a signal ("black and white" results) and to compensate for different exposure times of blots between replicate experiments, we set one high-intensity sample to "1" on each blot and calculated the other signals relative to this value, which gave good results.

We are now glad to provide the exact number of biological replicates, quantification data, mean values and standard deviations for all immunoblot experiments represented in Figures 1 to 6. In addition, we calculated p-values, which are also presented in supplementary Table 2. For Figure 4B, only quantitative data are given, because this experiment has been performed once (which was already stated in the legend of Figure 4 of the previous version of the manuscript). The reason is that it simply further confirms the result of Figure 4A. All other experiments were performed at least twice or more. However, we would like to emphasize that key results of this study were not only confirmed by biological replicates but also by experiments with different methods of gene inactivation (knockout, RNAi, inhibitors) and in different cell systems. We are optimistic that the manuscript is now adequate for publication in Nature Communications after our thorough quantitative and statistical analysis of immunoblots, which further supports our conclusions.

The authors, perhaps, are also disingenuous in their response to the reviews in that they do present measurements from their blots in one case, Figure 3C (without detailing how they obtained these numbers nor providing a statistical analysis of them).

Response: We did not intend to be disingenuous in our response and apologize if this was the perception of this reviewer. The reason could have been a misunderstanding. The measurements (numbers), which were shown in Figure 3C (and 3B) of the previous versions of this manuscript, represented phosphoimager data of radioactive substrate phosphorylations of the kinase assays displayed in the topmost panels as autoradiographs, but no immunoblots. For the sake of saving words, this was not explicitly mentioned in the Figure Legend, which made it probably difficult to understand what the Figures 3B and 3C in fact show. However, kinase assays and their quantification had been described in detail in the chapter "Immunocomplex Kinase Assays" in Methods. To make this point clearer to the reader we have revised the Figure Legend of Figure 3 and re-named the respective chapter in the Methods section: "Immunocomplex kinase assays and quantification". In addition, the kinase assay signals/topmost panels in Figures 3B and 3C have now been labeled with: IKK2 (resp. JNK1) activity, $^{32}\text{P}\sim\text{I}\kappa\text{B}\alpha$ (resp. $^{32}\text{P}\sim\text{GST-c-Jun}$). Quantification (numbers) of the displayed experiments has been removed from the figures. Instead, we have now performed quantitative analysis of the kinase assays underlying Figures 3B and 3C. The phosphoimager quantification results of three biological replicates and their statistical analysis are now presented in the new supplementary Table 3. They fully support the conclusions drawn from these experiments.

REVIEWERS' COMMENTS:

Reviewer #2 (Remarks to the Author):

The authors have addressed all of the concerns thoroughly and thoughtfully. This manuscript is now a lovely set of findings.